# GENERALIZED NEURAL COLLAPSE FOR A LARGE NUMBER OF CLASSES

## ABSTRACT

Neural collapse provides an elegant mathematical characterization of learned last layer representations (a.k.a. features) and classifier weights in deep classification models. Such results not only provide insights but also motivate new techniques for improving practical deep models. However, most of the existing empirical and theoretical studies in neural collapse focus on the case that the number of classes is small relative to the dimension of the feature space. This paper extends neural collapse to cases where the number of classes is much larger than the dimension of feature space, which broadly occurs for language models, retrieval systems, and face recognition applications. We show that the features and classifier exhibit a generalized neural collapse phenomenon, where the minimum one-vs-rest margins is maximized. We provide empirical study to verify the occurrence of generalized neural collapse in practical deep neural networks. Moreover, we provide theoretical study to show that the generalized neural collapse provably occurs under unconstrained feature model with spherical constraint, under certain technical conditions on feature dimension and number of classes.

## 1 INTRODUCTION

Over the past decade, deep learning algorithms have achieved remarkable progress across numerous machine learning tasks and have significantly enhanced the state-of-the-art in many practical applications ranging from computer vision to natural language processing and retrieval systems. Despite their tremendous success, a comprehensive understanding of the features learned from deep neural networks (DNNs) is still lacking. The recent work Papyan et al. (2020); Papyan (2020) has empirically uncovered an intriguing phenomenon regarding the last-layer features and classifier of DNNs, called *Neural Collapse* ($\mathcal{NC}$) that can be briefly summarized as the following characteristics:

- *Variability Collapse* ($\mathcal{NC}_1$): Within-class variability of features collapses to zero.

- *Convergence to Simplex ETF* ($\mathcal{NC}_2$): Class-mean features converge to a simplex Equiangular Tight Frame (ETF), achieving equal lengths, equal pair-wise angles, and maximal distance in the feature space.

- *Self-Duality* ($\mathcal{NC}_3$): Linear classifiers converge to class-mean features, up to a global rescaling.

Neural collapse provides a mathematically elegant characterization of learned representations or features in deep learning based classification models, independent of network architectures, dataset properties, and optimization algorithms. Building on the so-called *unconstrained feature model* (Mixon et al., 2020) or the *layer-peeled model* (Fang et al., 2021), subsequent research (Zhu et al., 2021; Lu & Steinerberger, 2020; Ji et al., 2021; Yaras et al.; Wojtowytsch et al., 2020; Ji et al.; Zhou et al.; Han et al.; Tirer & Bruna, 2022; Zhou et al., 2022a; Poggio & Liao, 2020; Thrampoulidis et al., 2022; Tirer et al., 2023; Nguyen et al., 2022) has provided theoretical evidence for the existence of the $\mathcal{NC}$ phenomenon when using a family of loss functions including cross-entropy (CE) loss, mean-square-error (MSE) loss and variants of CE loss. Theoretical results regarding $\mathcal{NC}$ not only contribute to a new understanding of the working of DNNs but also provide inspiration for developing new techniques to enhance their practical performance in various settings, such as imbalanced learning (Xie et al., 2023; Liu et al., 2023b), transfer learning (Galanti et al., 2022a; Li et al., 2022; Xie et al., 2022; Galanti et al., 2022b), continual learning (Yu et al., 2022; Yang et al., 2023), loss and architecture designs (Chan et al., 2022; Yu et al., 2020; Zhu et al., 2021), etc.

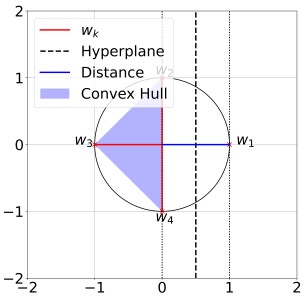
(a) One-vs-rest distance: Case 1

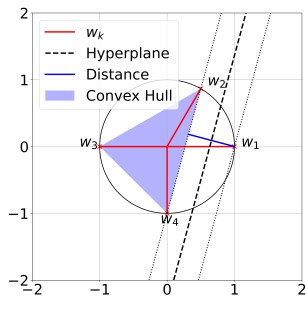
(b) One-vs-rest distance: Case 2

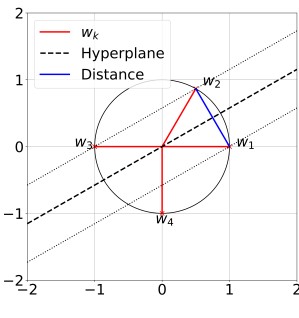
(c) One-vs-one distance

Figure 1: In Generalized Neural Collapse ($\mathcal{GNC}$), the optimal classifier weight $\{\boldsymbol{w}_k\}$ is a *Softmax Code* defined from maximizing the *one-vs-rest distance* (see Definition 2.1). *(a, b)* Illustration of the one-vs-rest distance using the example of $\boldsymbol{w}_1$-vs-$\{\boldsymbol{w}_2, \boldsymbol{w}_3, \boldsymbol{w}_4\}$ distance, under two configurations of $\{\boldsymbol{w}_k\}_{k=1}^4$ in a two-dimensional space. The distance in Case 1 is larger than that in Case 2. *(c)* Illustration of the *one-vs-one distance* used to define the Tammes problem (see Eq. (11)). We prove $\mathcal{GNC}$ under technical conditions on Softmax Code and Tammes problem (see Section 3).

However, most of the existing empirical and theoretical studies in $\mathcal{NC}$ focus on the case that the number of classes is small relative to the dimension of the feature space. Nevertheless, there are many cases in practice where the number of classes can be extremely large, such as

- Person identification (Deng et al., 2019), where each identity is regarded as one class.
- Language models (Devlin et al., 2018), where the number of classes equals the vocabulary size[1].
- Retrieval systems (Mitra et al., 2018), where each document in the dataset represents one class.
- Contrastive learning (Chen et al., 2020a), where each training data can be regarded as one class.

In such cases, it is usually infeasible to have a feature dimension commensurate with the number of classes due to computational and memory constraints. Therefore, it is crucial to develop a comprehensive understanding of the characteristics of learned features in such cases, particularly with the increasing use of web-scale datasets that have a vast number of classes.

**Contributions.** This paper studies the geometric properties of the learned last-layer features and the classifiers for cases where the number of classes can be arbitrarily large compared to the feature dimension. Motivated by the use of spherical constraints in learning with a large number of classes, such as person identification and contrastive learning, we consider networks trained with *spherical constraints* on the features and classifiers. Our contributions can be summarized as follows.

- **The Arrangement Problem: Generalizing $\mathcal{NC}$ to a Large Number of Classes.** In Section 2 we introduce the generalized $\mathcal{NC}$ ($\mathcal{GNC}$) for characterizing the last-layer features and classifier. In particular, $\mathcal{GNC}_1$ and $\mathcal{GNC}_3$ state the same as $\mathcal{NC}_1$ and $\mathcal{NC}_3$, respectively. $\mathcal{GNC}_2$ states that the classifier weight is a *Softmax Code*, which generalizes the notion of a simplex ETF and is defined as the collection of points on the unit hyper-sphere that maximizes the minimum one-vs-all distance (see Figure 1 (a,b) for an illustration). Empirically, we verify that the $\mathcal{GNC}$ approximately holds in practical DNNs trained with a small temperature in CE loss. Furthermore, we conduct theoretical study in Section 3 to show that under the unconstrained features model (UFM) (Mixon et al., 2020; Fang et al., 2021; Zhu et al., 2021) and with a vanishing temperature, the global solutions satisfy $\mathcal{GNC}$ under technical conditions on Softmax Code and solutions to the Tammes problem (Tammes, 1930), the latter defined as a collection of points on the unit hyper-sphere that maximizes the minimum one-vs-one distance (see Figure 1(c) for an illustration).

- **The Assignment Problem: Implicit Regularization of Class Semantic Similarity.** Unlike the simplex ETF (as in $\mathcal{NC}_2$) in which the distance between any pair of vectors is the same, not all pairs in a Softmax Code (as in $\mathcal{GNC}_2$) are of equal distant when the number of classes is greater than the feature space dimension. This leads to the "assignment" problem, i.e., the correspon-

---

[1]Language models are usually trained to classify a token (or a collection of them) that is either masked in the input (as in BERT (Devlin et al., 2018)), or the next one following the context (as in language modeling), or a span of masked tokens in the input (as in T5 (Raffel et al., 2020)), etc. In such cases, the number of classes is equal to the number of all possible tokens, i.e., the vocabulary size.

dence between the classes and the weights in a Softmax Code. In Section 4, we show empirically an implicit regularization effect by the semantic similarity of the classes, i.e., conceptually similar classes (e.g., Cat and Dog) are often assigned to closer classifier weights in Softmax Code, compared to those that are conceptually dissimilar (e.g., Cat and Truck). Moreover, such an implicit regularization is beneficial, i.e., enforcing other assignments produces inferior model quality.

- **Cost Reduction for Practical Network Training/Fine-tuning.** The universality of alignment between classifier weights and class means (i.e., $\mathcal{GNC}_3$) implies that training the classifier is unnecessary and the weight can be simply replaced by the class-mean features. Our experiments in Section 5 demonstrate that such a strategy achieves comparable performance to classical training methods, and even better out-of-distribution performance than classical fine-tuning methods with significantly reduced parameters.

**Related work.** The recent work Liu et al. (2023a) also introduces a notion of generalized $\mathcal{NC}$ for the case of large number of classes, which predicts equal-spaced features. However, their work focuses on networks trained with weight decay, for which empirical results in Appendix B.2 and Yaras et al. (2023) show to not produce equal-length and equal-spaced features for a relatively large number of classes. Due to limited space, we refer to Appendix B.2 for the detailed comparison between different geometric properties of the learned features and classifiers for the weight decay and spherical constraints formulations. Moreover, the work Liu et al. (2023a) relies on a specific choice of kernel function to describe the uniformity. Instead, we concretely define $\mathcal{GNC}_2$ through the softmax code. When preparing this submission, we notice a concurrent work Gao et al. (2023) that provides analysis for generalized $\mathcal{NC}$, but again for networks trained with weight decay. In addition, Gao et al. (2023) analyzes gradient flow for the corresponding UFM with a particular choice of weight decay, while our work studies the global optimality of the training problem. The work Zhou et al. (2022a) empirically shows that MSE loss is inferior to the CE loss when $K > d+1$, but no formal analysis is provided for CE loss. Finally, the global optimality of the UFM with spherical constraints has been studied in Lu & Steinerberger (2022); Yaras et al. (2023) but only for the cases $K \leq d + 1$ or $K \to \infty$.

## 2    GENERALIZED NEURAL COLLAPSE FOR A LARGE NUMBER OF CLASSES

In this section, we begin by providing a brief overview of DNNs and introducing notations used in this study in Section 2.1. We will also introduce the concept of the UFM which is used in theoretical study of the subsequent section. Next, we introduce the notion of *Softmax Code* for describing the distribution of a collection of points on the unit sphere, which prepares us to present a formal definition of *Generalized Neural Collapse* and empirical verification of its validity in Section 2.2.

### 2.1    BASICS CONCEPTS OF DNNS

A DNN classifier aims to learn a feature mapping $\phi_{\boldsymbol{\theta}}(\cdot) : \mathbb{R}^D \to \mathbb{R}^d$ with learnable parameters $\boldsymbol{\theta}$ that maps from input $\boldsymbol{x} \in \mathbb{R}^D$ to a deep representation called the feature $\phi_{\boldsymbol{\theta}}(\boldsymbol{x}) \in \mathbb{R}^d$, and a linear classifier $\boldsymbol{W} = [\boldsymbol{w}_1 \quad \boldsymbol{w}_2 \quad \cdots \quad \boldsymbol{w}_K] \in \mathbb{R}^{d \times K}$ such that the output (also known as the logits) $\Psi_{\boldsymbol{\Theta}}(\boldsymbol{x}) = \boldsymbol{W}^\top \phi_{\boldsymbol{\theta}}(\boldsymbol{x}) \in \mathbb{R}^K$ can make a correct prediction. Here, $\boldsymbol{\Theta} = \{\boldsymbol{\theta}, \boldsymbol{W}\}$ represents *all* the learnable parameters of the DNN.[2]

Given a balanced training set $\{(\boldsymbol{x}_{k,i}, \boldsymbol{y}_k)\}_{i \in [n], k \in [K]} \subseteq \mathbb{R}^D \times \mathbb{R}^K$, where $\boldsymbol{x}_{k,i}$ is the $i$-th sample in the $k$-th class and $\boldsymbol{y}_k$ is the corresponding one-hot label with all zero entries except for unity in the $k$-th entry, the network parameters $\boldsymbol{\Theta}$ are typically optimized by minimizing the following CE loss

$$\min_{\boldsymbol{\Theta}} \frac{1}{nK} \sum_{k=1}^{K} \sum_{i=1}^{n} \mathcal{L}_{\mathrm{CE}}\left(\Psi_{\boldsymbol{\Theta}}\left(\boldsymbol{x}_{k,i}\right), \boldsymbol{y}_k, \tau\right), \ \mathcal{L}_{\mathrm{CE}}\left(\boldsymbol{z}, \boldsymbol{y}_k, \tau\right) = -\log\left(\frac{\exp(z_k/\tau)}{\sum_{j=1}^{K} \exp(z_j/\tau)}\right). \quad (1)$$

In above, we assume that a spherical constraint is imposed on the feature and classifier weights and that the logit $z_k$ is divided by the temperature parameter $\tau$. This is a common practice when

---

[2] We ignore the bias term in the linear classifier since $(i)$ the bias term is used to compensate the global mean of the features and vanishes when the global mean is zero (Papyan et al., 2020; Zhu et al., 2021), $(ii)$ it is the default setting across a wide range of applications such as person identification (Wang et al., 2018b; Deng et al., 2019), contrastive learning (Chen et al., 2020a; He et al., 2020), etc.

dealing with a large number of classes (Wang et al., 2018b; Chang et al., 2019; Chen et al., 2020a). Specifically, we enforce $\{\boldsymbol{w}_k, \phi_{\boldsymbol{\Theta}}(\boldsymbol{x}_{k,i})\} \subseteq \mathbb{S}^{d-1} := \{\boldsymbol{a} \in \mathbb{R}^d : \|\boldsymbol{a}\|_2 = 1\}$ for all $i \in [n]$ and $k \in [K]$. An alternative regularization is weight decay on the model parameters $\boldsymbol{\Theta}$, the effect of which we study in Appendix B.

To simplify the notation, we denote the *oblique manifold* embedded in Euclidean space by $\mathcal{O}\mathrm{B}(d, K) := \left\{\boldsymbol{W} \in \mathbb{R}^{d \times K} \mid \boldsymbol{w}_k \in \mathbb{S}^{d-1}, \forall k \in [K]\right\}$. In addition, we denote the last-layer features by $\boldsymbol{h}_{k,i} := \phi_{\boldsymbol{\theta}}(\boldsymbol{x}_{k,i})$. We rewrite all the features in a matrix form as

$$\boldsymbol{H} := [\boldsymbol{H}_1 \quad \boldsymbol{H}_2 \quad \cdots \quad \boldsymbol{H}_K] \in \mathbb{R}^{d \times nK}, \text{with } \boldsymbol{H}_k := [\boldsymbol{h}_{k,1} \quad \cdots \quad \boldsymbol{h}_{k,n}] \in \mathbb{R}^{d \times n}.$$

Also we denote by $\overline{\boldsymbol{h}}_k := \frac{1}{n} \sum_{i=1}^n \boldsymbol{h}_{k,i}$ the class-mean feature for each class.

**Unconstrained Features Model (UFM).** The UFM (Mixon et al., 2020) or layer-peeled model (Fang et al., 2021), wherein the last-layer features are treated as free optimization variables, are widely used for theoretically understanding the $\mathcal{NC}$ phenomena. In this paper, we will consider the following UFM with a spherical constraint on classifier weights $\boldsymbol{W}$ and unconstrained features $\boldsymbol{H}$:

$$\min_{\boldsymbol{W}, \boldsymbol{H}} \frac{1}{nK} \sum_{k=1}^K \sum_{i=1}^n \mathcal{L}_{\mathrm{CE}}\left(\boldsymbol{W}^\top \boldsymbol{h}_{k,i}, \boldsymbol{y}_k, \tau\right) \quad \text{s.t.} \quad \boldsymbol{W} \in \mathcal{O}\mathrm{B}(d, K), \ \boldsymbol{H} \in \mathcal{O}\mathrm{B}(d, nK). \quad (2)$$

## 2.2 GENERALIZED NEURAL COLLAPSE

We start by introducing the notion of *softmax code* which will be used for describing $\mathcal{GNC}$.

**Definition 2.1** (Softmax Code). *Given positive integers $d$ and $K$, a softmax code is an arrangement of $K$ points on a unit sphere of $\mathbb{R}^d$ that maximizes the minimal distance between one point and the convex hull of the others:*

$$\max_{\boldsymbol{W} \in \mathcal{O}\mathrm{B}(d,K)} \rho_{\text{one-vs-rest}}(\boldsymbol{W}), \quad \text{where} \quad \rho_{\text{one-vs-rest}}(\boldsymbol{W}) \doteq \min_k \mathrm{dist}\left(\boldsymbol{w}_k, \{\boldsymbol{w}_j\}_{j \in [K] \setminus k}\right). \quad (3)$$

*In above, the distance between a point $\boldsymbol{v}$ and a set $\mathcal{W}$ is defined as $\mathrm{dist}(\boldsymbol{v}, \mathcal{W}) = \inf_{\boldsymbol{w} \in \mathrm{conv}(\mathcal{W})}\{\|\boldsymbol{v} - \boldsymbol{w}\|\}$, where $\mathrm{conv}(\cdot)$ denotes the convex hull of a set.*

We now extend $\mathcal{NC}$ to the *Generalized Neural Collapse* ($\mathcal{GNC}$) that captures the properties of the features and classifiers at the terminal phase of training. With a vanishing temperature (i.e., $\tau \to 0$), the last-layer features and classifier exhibit the following $\mathcal{GNC}$ phenomenon:

- *Variability Collapse* ($\mathcal{GNC}_1$). All features of the same class collapse to the corresponding class mean. Formally, as used in Papyan et al. (2020), the quantity $\mathcal{GNC}_1 \doteq \frac{1}{K} \mathrm{tr}\left(\boldsymbol{\Sigma}_W \boldsymbol{\Sigma}_B^\dagger\right) \to 0$, where $\boldsymbol{\Sigma}_B := \frac{1}{K} \sum_{k=1}^K \overline{\boldsymbol{h}}_k \overline{\boldsymbol{h}}_k^\top$ and $\boldsymbol{\Sigma}_W := \frac{1}{nK} \sum_{k=1}^k \sum_{i=1}^n \left(\boldsymbol{h}_{k,i} - \overline{\boldsymbol{h}}_k\right)\left(\boldsymbol{h}_{k,i} - \overline{\boldsymbol{h}}_k\right)^\top$ denote the between-class and within-class covariance matrices, respectively.

- *Softmax Codes* ($\mathcal{GNC}_2$). Classifier weights converge to the softmax code in definition 2.1. This property may be measured by $\mathcal{GNC}_2 \doteq \rho_{\text{one-vs-rest}}(\boldsymbol{W}) \to \max_{\boldsymbol{W} \in \mathcal{O}\mathrm{B}(d,K)} \rho_{\text{one-vs-rest}}(\boldsymbol{W})$.

- *Self-Duality* ($\mathcal{GNC}_3$). Linear classifiers converge to the class-mean features. Formally, this alignment can be measured by $\mathcal{GNC}_3 \doteq \frac{1}{K} \sum_{k=1}^K \left(1 - \boldsymbol{w}_k^\top \overline{\boldsymbol{h}}_k\right) \to 0$.

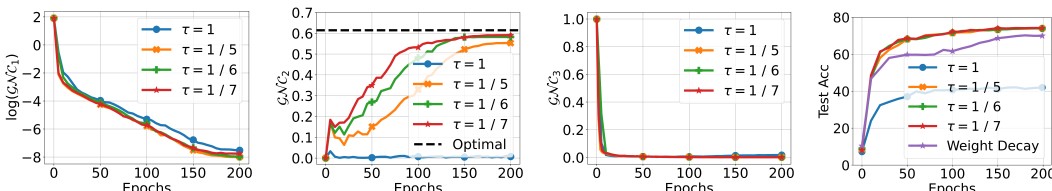

Figure 2: **Illustration of $\mathcal{GNC}$ and test accuracy across different temperatures** $\tau$ in training a ResNet18 on CIFAR100 with $d = 10$ and $K = 100$. "Optimal" in the second left figure refers to $\max_{\boldsymbol{W} \in \mathcal{O}\mathrm{B}(d,K)} \rho_{\text{one-vs-rest}}(\boldsymbol{W})$.

The main difference between $\mathcal{GNC}$ and $\mathcal{NC}$ lies in $\mathcal{GNC}_2 / \mathcal{NC}_2$, which describe the configuration of the classifier weight $\boldsymbol{W}$. In $\mathcal{NC}_2$, the classifier weights corresponding to different classes are

described as a simplex ETF, which is a configuration of vectors that have equal pair-wise distance and that distance is maximized. Such a configuration does not exist in general when the number of classes is large, i.e., $K > d + 1$. $\mathcal{GNC}_2$ introduces a new configuration described by the notion of softmax code. By Definition 2.1, a softmax code is a configuration where each vector is maximally separated from all the other points, measured by its distance to their convex hull. Such a definition is motivated from theoretical analysis (see Section 3). In particular, it reduces to simplex ETF when $K \leq d + 1$ (see Theorem 3.3).

**Interpretation of Softmax Code.** Softmax Code abides a max-distance interpretation. Specifically, consider the features $\{\boldsymbol{h}_{k,i}\}_{k \in [K], i \in [n]}$ from $n$ classes. In multi-class classification, one commonly used distance (or margin) measurement is the one-vs-rest (also called one-vs-all or one-vs-other) distance (Murphy, 2022), i.e., the distance of class $k$ vis-a-vis other classes. Noting that the distance between two classes is equivalent to the distance between the convex hulls of the data from each class (Murphy, 2022), the distance of class $k$ vis-a-vis other classes is given by $\text{dist}(\{\boldsymbol{h}_{k,i}\}_{i \in [n]}, \{\boldsymbol{h}_{k',i}\}_{k' \in [K] \backslash k, i \in [n]})$. From $\mathcal{GNC}_1$ and $\mathcal{GNC}_3$ we can rewrite the distance as

$$\text{dist}\big(\{\boldsymbol{h}_{k,i}\}_{i \in [n]}, \{\boldsymbol{h}_{k',i}\}_{k' \in [K] \backslash k, i \in [n]}\big) = \text{dist}\big(\overline{\boldsymbol{h}}_k, \{\overline{\boldsymbol{h}}_{k'}\}_{k' \in [K] \backslash k}\big) = \text{dist}\big(\boldsymbol{w}_k, \{\boldsymbol{w}_{k'}\}_{k' \in [K] \backslash k}\big).$$
(4)

By noticing that the rightmost term is minimized in a Softmax Code, it follows from $\mathcal{GNC}_2$ that the learned features satisfy that their one-vs-rest distance minimized over all classes $k \in [K]$ is maximized. In other words, measured by one-vs-rest distance, the learned features are are maximally separated. Finally, we mention that the separation of classes may be characterized by other measures of distance as well, such as the one-vs-one distance (also known as the sample margin in Cao et al. (2019); Zhou et al. (2022b)) which leads to the well-known Tammes problem, or the distances captured in the Thomson problems Thomson (1904); Hars. We will discuss this in Section 3.2.

**Experimental Verification of $\mathcal{GNC}$.** We verify the occurence of $\mathcal{GNC}$ by training a ResNet18 (He et al., 2016) for image classification on the CIFAR100 dataset (Krizhevsky, 2009), and report the results in Figure 2. To simulate the case of $K > d + 1$, we use a modified ResNet18 where the feature dimension is 10. From Figure 2, we can observe that both $\mathcal{GNC}_1$ and $\mathcal{GNC}_3$ converge to 0, and $\mathcal{GNC}_2$ converges towards the spherical code with relatively small temperature $\tau$. Additionally, selecting a small $\tau$ is not only necessary for achieving $\mathcal{GNC}$, but also for attaining high testing performance. Due to limited space, we present experimental details and other experiments with different architectures and datasets in Appendix B. In the next section, we provide a theoretical justification for $\mathcal{GNC}$ under UFM in (2).

## 3 THEORETICAL ANALYSIS OF GNC

In this section, we provide a theoretical analysis of $\mathcal{GNC}$ under the UFM in (2). We first show in Section 3.1 that under appropriate temperature parameters, the solution to (2) can be approximated by the solution to a "HardMax" problem, which is of a simpler form amenable for subsequent analysis. We then provide a theoretical analysis of $\mathcal{GNC}$ in Section 3.2, by first proving the optimal classifier forms a Softmax Code ($\mathcal{GNC}_2$), and then establishing $\mathcal{GNC}_1$ and $\mathcal{GNC}_3$ under technical conditions on Softmax Code and solutions to the Tammes problem. In addition, we provide insights for the design of feature dimension $d$ given a number of classes $K$ by analyzing the upper and lower bound for the one-vs-rest distance of a Softmax Code. All proofs can be found in Appendix C.

### 3.1 PREPARATION: THE ASYMPTOTIC CE LOSS

Due to the nature of the softmax function which blends the output vector, analyzing the CE loss can be difficult even for the unconstrained features model. The previous work Yaras et al. (2023) analyzing the case $K \leq d + 1$ relies on the simple structure of the global solutions, where the classifiers form a simplex ETF. However, this approach cannot be directly applied to the case $K > d + 1$ due to the absence of an informative characterization of the global solution. Motivated by the fact that the temperature $\tau$ is often selected as a small value ($\tau < 1$, e.g., $\tau = 1/30$ in Wang et al. (2018b)) in practical applications (Wang et al., 2018b; Chen et al., 2020a), we consider the case of $\tau \to 0$ where the CE loss (2) converges to the following "HardMax" problem:

$$\min_{\substack{\boldsymbol{W} \in \mathcal{OB}(d,K) \\ \boldsymbol{H} \in \mathcal{OB}(d,nK)}} \mathcal{L}_{\text{HardMax}}(\boldsymbol{W}, \boldsymbol{H}), \text{ where } \mathcal{L}_{\text{HardMax}}(\boldsymbol{W}, \boldsymbol{H}) \doteq \max_{k \in [K]} \max_{i \in [n]} \max_{k' \neq k} \langle \boldsymbol{w}_{k'} - \boldsymbol{w}_k, \boldsymbol{h}_{k,i} \rangle,$$
(5)

where $\langle \cdot, \cdot \rangle$ denotes the inner-product operator. More precisely, we have the following result.

**Lemma 3.1** (Convergence to the HardMax problem). *For any positive integers $K$ and $n$, we have*

$$\limsup_{\tau \to 0} \left( \underset{\substack{\boldsymbol{W} \in \mathcal{O}\mathrm{B}(d,K) \\ \boldsymbol{H} \in \mathcal{O}\mathrm{B}(d,nK)}}{\arg\min} \frac{1}{nK} \sum_{k=1}^{K} \sum_{i=1}^{n} \mathcal{L}_{CE} \left( \boldsymbol{W}^{\top} \boldsymbol{h}_{k,i}, \boldsymbol{y}_k, \tau \right) \right) \subseteq \underset{\substack{\boldsymbol{W} \in \mathcal{O}\mathrm{B}(d,K) \\ \boldsymbol{H} \in \mathcal{O}\mathrm{B}(d,nK)}}{\arg\min} \mathcal{L}_{HardMax}(\boldsymbol{W}, \boldsymbol{H}). \quad (6)$$

Our goal is not to replace CE with the HardMax function in practice. Instead, we will analyze the HardMax problem in (5) to gain insight into the global solutions and the $\mathcal{GNC}$ phenomenon.

## 3.2 Main Result: Theoretical Analysis of $\mathcal{GNC}$

$\mathcal{GNC}_2$ **and Softmax Code.** Our main result for $\mathcal{GNC}_2$ is the following.

**Theorem 3.2** ($\mathcal{GNC}_2$). *Let $(\boldsymbol{W}^\star, \boldsymbol{H}^\star)$ be an optimal solution to (5). Then, it holds that $\boldsymbol{W}^\star$ is a Softmax Code, i.e.,*

$$\boldsymbol{W}^\star = \underset{\boldsymbol{W} \in \mathcal{O}\mathrm{B}(d,K)}{\arg\max} \; \rho_{\text{one-vs-rest}}(\boldsymbol{W}). \quad (7)$$

$\mathcal{GNC}_2$ is described by the Softmax Code, which is defined from an optimization problem (see Definition 2.1). This optimization problem may not have a closed form solution in general. Nonetheless, the one-vs-rest distance that is used to define Softmax Code has a clear geometric meaning, making an intuitive interpretation of Softmax Code tractable. Specifically, maximizing the one-vs-rest distance results in the classifier weight vectors $\{\boldsymbol{w}_k^\star\}$ to be maximally distant. As shown in Figures 1a and 1b for a simple setting of four classes in a 2D plane, the weight vectors $\{\boldsymbol{w}_k\}$ that are uniformly distributed (and hence maximally distant) have a larger margin than the non-uniform case.

For certain choices of $(d, K)$ the Softmax Code bears a simple form.

**Theorem 3.3.** *For any positive integers $K$ and $d$, let $\boldsymbol{W}^\star \in \mathcal{O}\mathrm{B}(d,K)$ be a Softmax Code. Then,*

- $d = 2$: $\{\boldsymbol{w}_k^\star\}$ *is uniformly distributed on the unit circle, i.e.,* $\{\boldsymbol{w}_k^\star\} = \{\left( \cos(\frac{2\pi k}{K} + \alpha), \sin(\frac{2\pi k}{K} + \alpha) \right)\}$ *for some $\alpha$;*

- $K \leq d + 1$: $\{\boldsymbol{w}_k^\star\}$ *forms a simplex ETF, i.e.,* $\boldsymbol{W}^\star = \sqrt{\frac{K}{K-1}} \boldsymbol{P}(\boldsymbol{I}_K - \frac{1}{K}\boldsymbol{I}_K\boldsymbol{I}_K^\top)$ *for some orthonomal $\boldsymbol{P} \in \mathbb{R}^{d \times K}$;*

- $d + 1 < K \leq 2d$: $\rho_{\text{one-vs-rest}}(\boldsymbol{W}^\star) = 1$ *which can be achieved when $\{\boldsymbol{w}_k^\star\}$ are a subset of vertices of a cross-polytope[3];*

For the cases of $K \leq d + 1$, the optimal $\boldsymbol{W}^\star$ from Theorem 3.3 is the same as that of Lu & Steinerberger (2022). However, Theorem 3.3 is an analysis of the HardMax loss while Lu & Steinerberger (2022) analyzed the CE loss.

$\mathcal{GNC}_1$ **and Within-class Variability Collapse.** To establish the within-class variability collapse property, we require a technical condition associated with the Softmax Code. Recall that Softmax Codes are those that maximize the *minimum* one-vs-rest distance over all classes. We introduce *rattlers*, which are classes that do not attain such a *minimum*.

**Definition 3.4** (Rattler of Softmax Code). *Given positive integers $d$ and $K$, a rattler associated with a Softmax Code $\boldsymbol{W}^{SC} \in \mathcal{O}\mathrm{B}(d,K)$ is an index $k_{rattler} \in [K]$ for which*

$$\min_{k \in [K]} \text{dist}(\boldsymbol{w}_k^{SC}, \{\boldsymbol{w}_j^{SC}\}_{j \in [K] \setminus k}) \neq \text{dist}(\boldsymbol{w}_{k_{rattler}}^{SC}, \{\boldsymbol{w}_j^{SC}\}_{j \in [K] \setminus k_{rattler}}). \quad (8)$$

In other words, rattlers are points in a Softmax Code with no neighbors at the minimum one-to-rest distance. This notion is borrowed from the literature of the *Tammes Problem* (Cohn, 2022; Wang, 2009), which we will soon discuss in more detail[4].

We are now ready to present the main results for $\mathcal{GNC}_1$.

---

[3]Indeed, any sphere code $\boldsymbol{W}$ that achieves equality in Rankin's orthoplex bound (Fickus et al., 2017) $\max_{k \neq j} \langle \boldsymbol{w}_k, \boldsymbol{w}_j \rangle \geq 0$ is a softmax code.

[4]The occurrence of rattlers is rare: Among the 182 pairs of $(d, K)$ for which the solution to Tammes problem is known, only 31 have rattlers (Cohn, 2022). This has excluded the cases of $d = 2$ or $K \leq 2d$ where there is no rattler. The occurrence of ratter in Softmax Code may be rare as well.

**Theorem 3.5** ($\mathcal{GNC}_1$)**.** *Let* $(\boldsymbol{W}^\star, \boldsymbol{H}^\star)$ *be an optimal solution to* (5)*. For all* $k$ *that is not a rattler of* $\boldsymbol{W}^\star$*, it holds that*

$$\overline{\boldsymbol{h}}_k^\star \doteq \boldsymbol{h}_{k,1}^\star = \cdots = \boldsymbol{h}_{k,n}^\star = \mathcal{P}_{\mathbb{S}^{d-1}} \left( \boldsymbol{w}_k^\star - \mathcal{P}_{\{\boldsymbol{w}_j^\star\}_{j \in [K] \backslash k}}(\boldsymbol{w}_k^\star) \right), \tag{9}$$

*where* $\mathcal{P}_\mathcal{W}(\boldsymbol{v}) \doteq \arg\min_{\boldsymbol{w} \in \mathrm{conv}(\mathcal{W})}\{\|\boldsymbol{v} - \boldsymbol{w}\|_2\}$ *denotes the projection of* $\boldsymbol{v}$ *on* $\mathrm{conv}(\mathcal{W})$*.*

The following result shows that the requirement in the Theorem 3.5 that $k$ is not a rattler is satisfied in certain cases.

**Theorem 3.6.** *If* $d = 2$*, or* $K \leq d + 1$*, Softmax Code has no rattler for all classes.*

$\mathcal{GNC}_3$ **and Self-Duality.** To motivate our technical conditions for establishing self-duality, assume that any optimal solution $(\boldsymbol{W}^\star, \boldsymbol{H}^\star)$ to (5) satisfies self-duality as well as $\mathcal{GNC}_1$. This implies that

$$\underset{\boldsymbol{W} \in \mathcal{OB}(d,K), \boldsymbol{H} \in \mathcal{OB}(d,nK)}{\arg\min} \mathcal{L}_{\mathrm{HardMax}}(\boldsymbol{W}, \boldsymbol{H}) = \underset{\boldsymbol{W} \in \mathcal{OB}(d,nK)}{\arg\min} \max_{k \in [K]} \max_{i \in [n]} \max_{k' \neq k} \langle \boldsymbol{w}_{k'} - \boldsymbol{w}_k, \boldsymbol{w}_k \rangle. \tag{10}$$

After simplification we may rewrite the optimization problem on the right hand side equivalently as:

$$\max_{\boldsymbol{W} \in \mathcal{OB}(d,K)} \rho_{\text{one-vs-one}}(\boldsymbol{W}), \quad \text{where} \quad \rho_{\text{one-vs-one}}(\boldsymbol{W}) \doteq \min_{k \in [K]} \min_{k' \neq k} \mathrm{dist}(\boldsymbol{w}_k, \boldsymbol{w}_{k'}). \tag{11}$$

Eq. (11) is the well-known *Tammes problem*. Geometrically, the problem asks for a distribution of $K$ points on the unit sphere of $\mathbb{R}^d$ so that the minimum distance between any pair of points is maximized. The Tammes problem is unsolved in general, except for certain pairs of $(K, d)$.

Both the Tammes problem and the Softmax Code are problems of arranging points to be maximally separated on the unit sphere, with their difference being the specific measures of separation. Comparing (11) and (3), the Tammes problem maximizes for all $k \in [K]$ the *one-vs-one distance*, i.e., $\min_{k' \neq k} \mathrm{dist}(\boldsymbol{w}_k, \boldsymbol{w}_{k'})$, whereas the Softmax Code maximizes the minimum *one-vs-rest distance*, i.e., $\mathrm{dist}(\boldsymbol{w}_k, \{\boldsymbol{w}_j\}_{j \in [K] \backslash k})$. Both one-vs-one distance and one-vs-rest distances characterize the separation of the weight vector $\boldsymbol{w}_k$ from $\{\boldsymbol{w}_j\}_{j \in [K] \backslash k}$. As illustrated in Figure 1, taking $k = 1$, the former is the distance between $\boldsymbol{w}_1$ and its closest point in the set $\{\boldsymbol{w}_2, \boldsymbol{w}_3, \boldsymbol{w}_4\}$, in this case $\boldsymbol{w}_2$ (see Figure 1c), whereas the later captures the minimal distance from $\boldsymbol{w}_1$ to the convex hull of the rest vectors $\{\boldsymbol{w}_1, \boldsymbol{w}_2, \boldsymbol{w}_3\}$ (see Figure 1b).

Since the Tammes problem can be derived from the self-duality constraint on the HardMax problem, it may not be surprising that the Tammes problem can be used to describe a condition for establishing self-duality. Specifically, we have the following result.

**Theorem 3.7** ($\mathcal{GNC}_3$)**.** *For any* $K, d$ *such that both Tammes problem and Softmax Code have no rattler, the following two statements are equivalent:*

- *Any optimal solution* $(\boldsymbol{W}^\star, \boldsymbol{H}^\star)$ *to* (5) *satisfies* $\boldsymbol{h}_{k,i}^\star = \boldsymbol{w}_k^*, \forall i \in [n], \forall k \in [K]$*;*

- *The Tammes problem and the Softmax codes are equivalent, i.e.,*

$$\underset{\boldsymbol{W} \in \mathcal{OB}(d,K)}{\arg\max} \rho_{\text{one-vs-rest}}(\boldsymbol{W}) = \underset{\boldsymbol{W} \in \mathcal{OB}(d,K)}{\arg\max} \rho_{\text{one-vs-one}}(\boldsymbol{W}). \tag{12}$$

In words, Theorem 3.7 states that $\mathcal{GNC}_3$ holds if and only if the Tammes problem in (11) and the Softmax codes are equivalent. As both the Tammes problem and Softmax Code maximize separation between one vector and the others, though their notions of separation are different, we conjecture that they are equivalent and share the same optimal solutions. We prove this conjecture for some special cases and leave the study for the general case as future work[5].

**Theorem 3.8.** *If* $d = 2$*, or* $K \leq d + 1$*, the Tammes problem and the Softmax codes are equivalent.*

### 3.3 INSIGHTS FOR CHOOSING FEATURE DIMENSION $d$ GIVEN CLASS NUMBER $K$

Given a class number $K$, how does the choice of feature dimension $d$ affect the model performance? Intuitively, smaller $d$ reduces the separability between classes in a Softmax Code. We define this rigorously by providing bounds for the one-vs-rest distance of a Softmax Code based on $d$ and $K$.

---

[5]We numerically verify the equivalence for all the cases with $d \leq 100$ in Table 1 of Cohn & Kumar (2007).

**Theorem 3.9.** *Assuming $K \geq \sqrt{2\pi\sqrt{e}d}$ and letting $\Gamma(\cdot)$ denote the Gamma function, we have*

$$\frac{1}{2}\left[\frac{\sqrt{\pi}}{K}\frac{\Gamma\left(\frac{d+1}{2}\right)}{\Gamma\left(\frac{d}{2}+1\right)}\right]^{\frac{2}{d-1}} \leq \max_{\boldsymbol{W}\in\mathcal{OB}(d,K)} \rho_{\text{one-vs-rest}}(\boldsymbol{W}) \leq 2\left[\frac{2\sqrt{\pi}}{K}\frac{\Gamma\left(\frac{d+1}{2}\right)}{\Gamma\left(\frac{d}{2}\right)}\right]^{\frac{1}{d-1}}. \tag{13}$$

The bounds characterize the separability for $K$ classes in $d$-dimensional space. Given the number of classes $K$ and desired margin $\rho$, the minimal feature dimension is roughly an order of $\log(K^2/\rho)$, showing classes separate easily in higher dimensions. This also provides a justification for applications like face classification and self-supervised learning, where the number of classes (e.g., millions of classes) could be significantly larger than the dimensionality of the features (e.g., $d = 512$).

By conducting experiments on ResNet-50 with varying feature dimensions for ImageNet classification, we further corroborate the relationship between feature dimension and network performance in Figure 3. First, we observe that the curve of the optimal distance is closely aligned with the curve of testing performance, indicating a strong correlation between distance and testing accuracy. Moreover, both the distance and performance curves exhibit a slow (exponential) decrease as the feature dimension $d$ decreases, which is consistent with the bounds in Theorem 3.9.

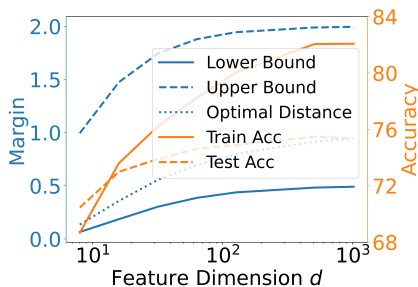

Figure 3: Effect of feature dimension $d$ on (Left $y$-axis): $\rho_{\text{one-vs-rest}}(\boldsymbol{W}^{\star})$ and its upper/lower bounds (in Theorem 3.9), and (Right $y$-axis): training and test accuracies for ResNet-50 on ImageNet.

## 4 THE ASSIGNMENT PROBLEM: AN EMPIRICAL STUDY

Unlike the case $d \geq K - 1$ where the optimal classifier (simplex ETF) has equal angles between any pair of the classifier weights, when $d < K - 1$, not all pairs of classifier weights are equally distant with the optimal $\boldsymbol{W}$ (Softmax Code) predicted in Theorem 3.2. Consequently, this leads to a "class assignment" problem. To illustrate this, we train a ResNet18 network with $d = 2$ on four classes {Automobile, Cat, Dog, Truck} from CIFAR10 dataset that are selected due to their clear semantic similarity and discrepancy. In this case, according to Theorem 3.3, the optimal classifiers are given by $[1, 0], [-1, 0], [0, 1], [0, -1]$, up to a rotation. Consequently, there are three distinct class assignments, as illustrated in Figures 4b to 4d.

When doing standard training, the classifier consistently converges to the case where Cat and Dog are closer together across 5 different trials; Figure 4a shows the learned features (dots) and classifier weights (arrows) in one of such trials. This demonstrates the implicit algorithmic regularization in training DNNs, which naturally attracts (semantically) similar classes and separates dissimilar ones.

We also conduct experiments with the classifier fixed to be one of the three arrangements, and present the results in Figures 4b to 4d. Among them, we observe that the case where Cat and Dog are far apart achieves a testing accuracy of $89.95\%$, lower than the other two cases with accuracies of $91.90\%$ and $92.13\%$. This demonstrates the important role of class assignment to the generalization of DNNs, and that the implicit bias of the learned classifier is benign, i.e., leads to a more generalizable solutions. A comprehensive study of this phenomenon is deferred to future work.

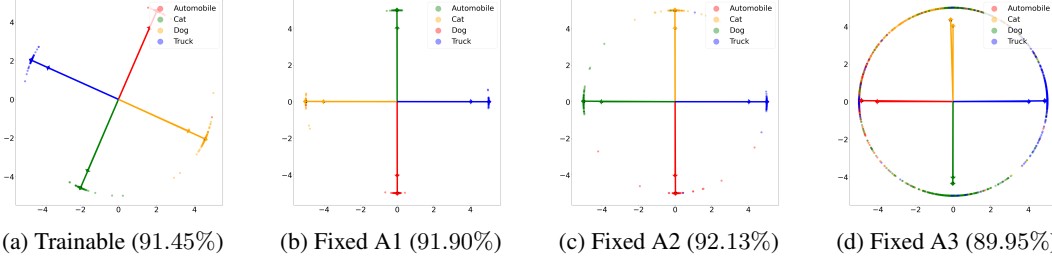

(a) Trainable (91.45%)    (b) Fixed A1 (91.90%)    (c) Fixed A2 (92.13%)    (d) Fixed A3 (89.95%)

Figure 4: **Assignment of classes to classifier weights** for a ResNet18 with 2-dimensional feature space trained on the 4 classes {Automobile, Cat, Dog, Truck} from CIFAR10. *(a)* Learned classifier. *(b-d)* Classifiers fixed to be three different assignments. Test accuracy is reported in the bracket.

## 5 IMPLICATIONS FOR PRACTICAL NETWORK TRAINING/FINE-TUNING

Since the classifier always converges to a simplex ETF when $K \leq d + 1$, prior work proposes to fix the classifier as a simplex ETF for reducing training cost (Zhu et al., 2021) and handling imbalance dataset (Yang et al., 2022). When $K > d + 1$, the optimal classifier is also known to be a Softmax Code according to $\mathcal{GNC}_2$. However, the same method as in prior work may become sub-optimal due to the class assignment problem (see Section 4). To address this, we introduce the method of class-mean features (CMF) classifiers, where the classifier weights are set to be the exponential moving average of the mini-batch class-mean features during the training process. This approach is motivated from $\mathcal{GNC}_3$ which states that the optimal classifier converges to the class-mean features. We explain the detail of CMF in Appendix B. As in prior work, CMF can reduce trainable parameters as well. For instance, it can reduce 30.91% of total parameters in a ResNet18 for BUPT-CBFace-50 dataset (Zhang & Deng, 2020). Here, we compare CMF with the standard training where the classifier is learned together with the feature mapping, in both training from scratch and fine-tuning.

**Training from Scratch.** We train a ResNet18 on CIFAR100 by using a learnable classifier or the CMF classifier. The learning curves in Figure 5 indicate that the approach with CMF classifier achieves comparable performance to the classical training protocols.

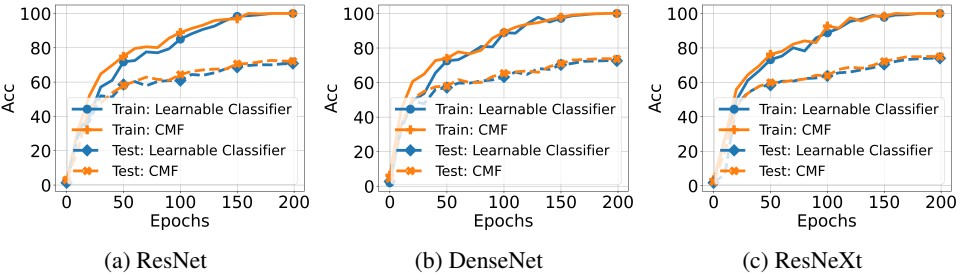

|   (a) ResNet   |   (b) DenseNet   |   (c) ResNeXt   |

Figure 5: **Comparison of the learning curves (training and testing accuracies) with learned classifiers vs. CMF classifiers** trained with various networks on CIFAR100 dataset and $d = 10$.

**Fine-tuning.** To verify the effectiveness of the CMF classifiers on fine-tuning, we follow the setting in Kumar et al. (2022) to measure the performance of the fine-tuned model on both in-distribution (ID) task (i.e., CIFAR10 Krizhevsky (2009)) and OOD task (STL10 Coates et al. (2011)). We compare the standard approach that fine-tunes both the classifier (randomly initialized) and the pre-trained feature mapping with our approach (using the CMF classifier). Our experiments show that the approach with CMF classifier achieves slightly better ID accuracy (98.00% VS 97.00%) and a better OOD performance (90.67% VS 87.42%). The improvement of OOD performance stems from the ability to align the classifier with the class-means through the entire process, which better preserves the OOD property of the pre-trained model. Our approach also simplifies the two-stage approach of linearly probing and subsequent full fine-tuning in Kumar et al. (2022).

## 6 CONCLUSION

In this work, we have introduced generalized neural collapse ($\mathcal{GNC}$) for characterizing learned last-layer features and classifiers in DNNs under an arbitrary number of classes and feature dimensions. We empirically validate the $\mathcal{GNC}$ phenomenon on practical DNNs that are trained with a small temperature in the CE loss and subject to spherical constraints on the features and classifiers. Building upon the unconstrained features model we have proven that $\mathcal{GNC}$ holds under certain technical conditions. $\mathcal{GNC}$ could offer valuable insights for the design, training, and generalization of DNNs. For example, the minimal one-vs-rest distance provides implications for designing feature dimensions when dealing with a large number of classes. Additionally, we have leveraged $\mathcal{GNC}$ to enhance training efficiency and fine-tuning performance by fixing the classifier as class-mean features. Further exploration of $\mathcal{GNC}$ in other scenarios, such as imbalanced learning, is left for future work. It is also of interest to further study the problem of optimally assigning classifiers from Softmax Code for each class, which could shed light on developing techniques for better classification performance.

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
