# Appendix

The organization of the appendix is as follows. Firstly, we introduce some useful results that are utilized throughout the appendices, and discuss related works on using CE loss with spherical constraints. In Appendix B, we offer comprehensive information regarding the datasets and computational resources for each figure, along with presenting additional experimental results on practical datasets. Lastly, in Appendix C, we provide the theoretical proofs for all the theorems mentioned in Section 3.

## A  BASIC AND COMMON PRACTICE OF USING SPHERICAL CONSTRAINTS.

### A.1  USEFUL RESULTS

**Lemma A.1** (LogSumExp). *Let denote $\boldsymbol{x} = [x_1, x_2, \cdots, x_n] \in \mathcal{R}^n$ and the LogSumExp function $LSE(\boldsymbol{x}, \tau) = \log\left(\sum_{i=1}^n \exp(x_i/\tau)\right)$. With $\tau \to 0$, we have*

$$\tau LSE(\boldsymbol{x}, \tau) \to \max_i x_i$$

*In other words, the LogSumExp function is a smooth approximation to the maximum function.*

*Proof of Lemma A.1.* According to the definition of the LogSumExp function, we have

$$\max_i x_i \le \tau \mathrm{LSE}(\boldsymbol{x}, \tau) \le \max_i x_i + \tau \log n$$

where the first inequality is strict unless $n = 1$ and the second inequality is strict unless all arguments are equal $x_1 = x_2 = \cdots = x_n$. Therefore, with $\tau \to 0$, $\tau \mathrm{LSE}(\boldsymbol{x}, \tau) \to \max_i x_i$.  □

**Theorem A.2** (Carathéodory's theorem Carathéodory (1911)). *Given $\boldsymbol{w}_1, \ldots, \boldsymbol{w}_K \in \mathbb{R}^d$, if a point $\boldsymbol{v}$ lies in the convex hull $\mathrm{conv}(\{\boldsymbol{w}_1, \ldots, \boldsymbol{w}_K\})$ of $\{\boldsymbol{w}_1, \ldots, \boldsymbol{w}_K\}$, then $\boldsymbol{v}$ resides in the convex hull of at most $d + 1$ of the points in $\{\boldsymbol{w}_1, \ldots, \boldsymbol{w}_K\}$.*

### A.2  COMMON PRACTICE OF USING SPHERICAL CONSTRAINTS

In cases where the number of classes is larger than the feature dimensions, it is common practice to use a modified version of CE loss (1) with normalization and a small temperature $\tau$. This approach is prevalent in face recognition, information retrieval and self-supervised contrastive learning areas. Below we present some representative works.

- **Face recognition.** AdditiveFace Wang et al. (2018a), ArcFace Deng et al. (2019), NormFace Wang et al. (2017) and CosFace Wang et al. (2018b) use the cross entropy loss with the feature and weight normalization and small temperature parameter. They observe that the feature and weight normalization are necessary since this practice strengthens the cosine constraint and improves the model's ability to distinguish between different face classes. For the temperature values, AdditiveFace Wang et al. (2018a) uses the temperature $\tau = 1/30$ (in the first sentence of page 3); ArcFace Deng et al. (2019) sets $\tau = 1/64$ (in the second paragraph of experiment setting in section 4.2); In the NormFace Wang et al. (2017), the authors conduct comprehensive experiments for the effect of temperature across different datasets. For example, the superior performance is chosen at temperature $\tau = 1/40$ on the LFW dataset in figure 8, which is also better than CE loss without any normalization. The CosFace Wang et al. (2018b) sets the temperature parameter as $\tau = 1/64$ ("training part" of section 4.1) for CASIA-WebFace and $\tau = 1/30$ also performs well in the practical implementation [link].
- **Information retrieval.** Yi et al. (2019) also uses the modified cross entropy loss when training dual encoders (the last paragraph in section 3) to learn the representations of query and candidates. They observe that setting a small temperature value from $\tau = 0.05$ to $0.07$ results in the best performance in table 1. Similarly, Lindgren et al. (2021) also uses the modified cross entropy loss when training dual encoders in Equation 1 & 2 of section 2.2. It uses a temperature value of $\tau = 0.05$ for document retrieval (see the "set up the cache loss" in [link]).

- **Contrastive learning.** SimCLR Chen et al. (2020a) treats each data sample as a class and computes the cosine similarity between pairs of positive and negative examples using the normalized inner product of features. It chooses the inverse temperature between 10 and 20 (which is the inverse of our $\tau$) in table 5 to obtain the best performance. Similarly, MoCo He et al. (2020) sets temperature as $\tau = 0.07$ according to the "technical details" part of section 3.3. In multi-domains constrative learning, ConVIRT Zhang et al. (2022) uses normalization and temperature for the CE loss to learn contrastive representations between medical image and text (see equation 2 & 3) and chooses $\tau = 0.01$ (in table 4). CLIP Radford et al. (2021) model applies normalization to both image and text embeddings to ensure consistent scales(see figure 3), and it initializes the temperature $\tau = 0.07$ (in section 2.5 of page 5).

## B EXPERIMENT

In this section, we begin by providing additional details regarding the datasets and the computational resources utilized in the paper. Specifically, CIFAR10, CIFAR100, and BUPT-CBFace datasets are publicly available for academic purposes under the MIT license. Additionally, all experiments were conducted on 4xV100 GPU with 32G memory. Furthermore, we present supplementary experiments and implementation specifics for each figure.

### B.1 IMPLEMENTATION DETAILS

We present implementation details for results in the paper.

**Results in Figure 2.** To illustrate the occurrence of the $\mathcal{GNC}$ phenomenon in practical multi-class classification problems, we trained a ResNet18 network (He et al., 2016) on the CIFAR100 dataset (Krizhevsky, 2009) using CE loss with varying temperature parameters. In this experiment, we set the dimensions of the last-layer features to 10. This is achieved by setting the number of channels in the second convolutional layer of the last residual block before the classifier layer to be 10. Prior to training, we applied standard preprocessing techniques, which involved normalizing the images (channel-wise) using their mean and standard deviation. We also employed standard data augmentation methods. For optimization, we utilized SGD with a momentum of 0.9 and an initial learning rate of 0.1, which decayed according to the CosineAnnealing over a span of 200 epochs.

The optimal margin (represented by the dotted line) in the second left subfigure is obtained from numerical optimization of the Softmax Code problem. In this optimization process, we used an initial learning rate of 0.1 and decreased it by a factor of 10 every 1000 iterations, for a total of 5000 iterations.

**Results in Figure 4.** To demonstrate the implicit algorithmic regularization in training DNNs, we train a ResNet18 network on four classes Automobile, Cat, Dog, Truck from CIFAR10 dataset. We set the dimensions of the last-layer features to 2 so that the learned features can be visualized, and we used a temperature parameter of 0.05. We optimized the networks for a total of 800 epochs using SGD with a momentum of 0.9 and an initial learning rate of 0.1, which is decreased by a factor of 10 every 200 epochs.

**Results in Figure 5.** To assess the effectiveness of the proposed CMF (Class Mean Feature) method, we utilized the ResNet18 architecture (He et al., 2015) as the feature encoder on the CIFAR100 dataset (Krizhevsky, 2009), where we set the feature dimension to $d = 20$ and the temperature parameter to $\tau = 0.1$. Since the model can only access a mini-batch of the dataset for each iteration, it becomes computationally prohibitive to calculate the class-mean features of the entire dataset. As a solution, we updated the classifier by employing the exponential moving average of the feature class mean, represented as $\boldsymbol{W}^{(t+1)} \leftarrow \beta \boldsymbol{W}^{(t)} + (1 - \beta)\overline{\boldsymbol{H}}^{(t)}$. Here, $\overline{\boldsymbol{H}}^{(t)} \in \mathcal{R}^{K \times d}$ denotes the class mean feature of the mini-batch in iteration $t$, $\boldsymbol{W} \in \mathcal{R}^{K \times d}$ represents the classifier weights, and $\beta \in [0, 1)$ denotes the momentum coefficient (in our experiment, $\beta = 0.9$). We applied the same data preprocessing and augmentation techniques mentioned previously and employed an initial learning rate of 0.1 with the CosineAnnealing scheduler.

**Results in the fine-tuning experiment of Section 5.** We fine-tune a pretrained ResNet50 on MoCo v2 (Chen et al., 2020b) on CIFAR10 (Krizhevsky, 2009). The CIFAR10 test dataset was chosen as the in-distribution (ID) task, while the STL10 dataset (Coates et al., 2011) served as the out-of-distribution (OOD) task. To preprocess the dataset, we resized the images to $224 \times 224$ using BICUBIC interpolation and normalized them by their mean and standard deviation. Since there is no "monkey" class in CIFAR10 dataset, we remove the "monkey" class in the STL10 dataset. Additionally, we reassign the labels of CIFAR10 datasets to the STL10 dataset in order to match the classifier output. For optimization, we employed the Adam optimizer with a learning rate of $1e-5$ and utilized the CosineAnnealing scheduler. The models are fine-tuned for 5 epochs with the batch size of 100. We reported the best ID testing accuracy achieved during the fine-tuning process and also provided the corresponding OOD accuracy.

**Results in Figure B.1.** To visualize the different structures learned under weight decay or spherical constraint, we conducted experiments in both practical and unconstrained feature model settings. In the practical setting, we trained a ResNet18 network (He et al., 2016) on the first 30 classes of the CIFAR100 dataset (Krizhevsky, 2009) using either weight decay or spherical constraint with the cross-entropy (CE) loss. For visualization purposes, we set the dimensions of the last-layer features to 2, and for the spherical constraint, we used a temperature parameter of 70. Prior to training, we applied standard preprocessing techniques, which included normalizing the images (channel-wise) using their mean and standard deviation. Additionally, we employed standard data augmentation methods. We optimized the networks for a total of 1600 epochs using SGD with a momentum of 0.9 and an initial learning rate of 0.1, which decreased by a factor of 10 every 700 epochs. For the unconstrained feature model setting, we considered only one sample per class and treated the feature (class-mean features) and weights as free optimization variables. In this case, we initialized the learning rate at 0.1 and decreased it by a factor of 10 every 1000 iterations, for totaling 5000 iterations. The temperature parameter was set to 0.02 for the spherical constraint.

**Results in Figure B.6.** To demonstrate the implicit algorithmic regularization in training DNNs, we train a ResNet18 network on four classes Automobile, Cat, Deer, Dog, Horse, Truck from CIFAR10 dataset. We set the dimensions of the last-layer features to 2 so that the learned features can be visualized, and we used a temperature parameter of 0.2. We optimized the networks for a total of 600 epochs using SGD with a momentum of 0.9 and an initial learning rate of 0.1, which is decreased by a factor of 10 every 200 epochs.

## B.2 EFFECT OF REGULARIZATION: SPHERICAL CONSTRAINT VS. WEIGHT DECAY

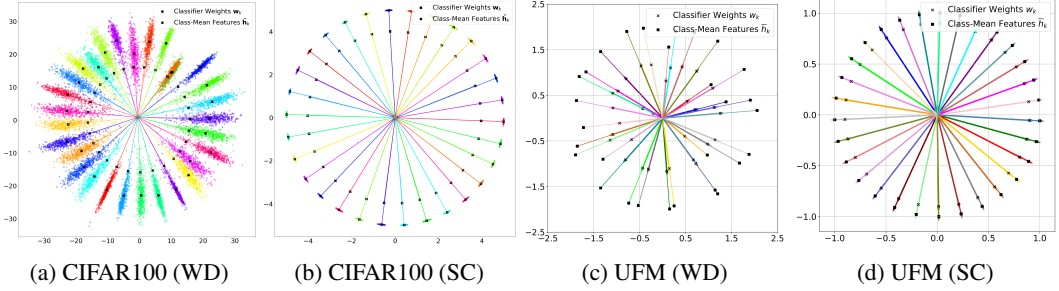

(a) CIFAR100 (WD)  (b) CIFAR100 (SC)  (c) UFM (WD)  (d) UFM (SC)

Figure B.1: **Comparison of classifier weights and last-layer features with weight decay (WD) vs spherical constraint (SC) on features and classifiers.** (a) vs (b): Results with a ResNet18 trained on CFIAR100. (c) vs (d): Results with the unconstrained feature model in (B.1) and (2), respectively. In all settings, we set $K = 30$ and $d = 2$ for visualization; the first $K = 30$ classes from CIFAR100 dataset are used to train ResNet18.

We study the features and classifiers of networks trained with weight decay for the case $K > d + 1$, and compare with those obtained with spherical constraint. For the purpose of visualization, we train a ResNet18 with $d = 2$ on the first $K = 30$ classes of CIFAR100 and display the learned features and classifiers in Figure B.1(a). Below we summarize observations from Figure B.1(a).

- **Non-equal length and non-uniform distribution.** The vectors of class-mean features $\{\overline{h}_k\}$ *do not* have equal length, and also appear to be *not* equally spaced when normalized to the unit sphere.
- **Self-duality only in direction.** The classifiers point towards the same direction as their corresponding class-mean features, but they have different lengths, and there exists no global scaling to exactly align them. As shown in Figure B.2, the ratios between the lengths of classifier weights and class-mean features vary across different classes. Therefore, there exists no global scaling to align them exactly.

In contrast, using spherical constraint produces equal-length and uniformly distributed features, as well as aligned classifier and classifier weights not only in direction but also in length, see Figure B.1 (b). Such a result is aligned with $\mathcal{GNC}$. This observation may explain the common practice of using feature normalization in applications with an extremely large number of classes (Chen & He, 2021; Wang et al., 2018b), and justify our study of networks trained with sphere constraints in (2) rather than weight decay as in Liu et al. (2023a).

We note that the discrepancy between the approaches using weight decay and spherical constraints is not due to the insufficient expressiveness of the networks. In fact, we also observe different performances in the unconstrained feature model with spherical constraints as in (2) and with the following regularized form (Zhu et al., 2021; Mixon et al., 2020; Tirer & Bruna, 2022; Zhou et al., 2022a):

$$\min_{\boldsymbol{W},\boldsymbol{H}} \frac{1}{nK} \sum_{k=1}^{K} \sum_{i=1}^{n} \mathcal{L}_{\text{CE}} \left( \boldsymbol{W}^\top \boldsymbol{h}_{k,i}, \boldsymbol{y}_k, \tau \right) + \frac{\lambda}{2} (\|\boldsymbol{W}\|_F^2 + \|\boldsymbol{H}\|_F^2), \tag{B.1}$$

where $\lambda$ represents the weight decay parameters. We observe similar phenomena in Figure B.1(c, d) as in Figure B.1(a, b) that the weight decay formulation results in features with non-equal lengths, non-uniform distribution, and different lengths than the classifiers. In the next section, we provide a theoretical justification for $\mathcal{GNC}$ under the UFM for Problem (2).

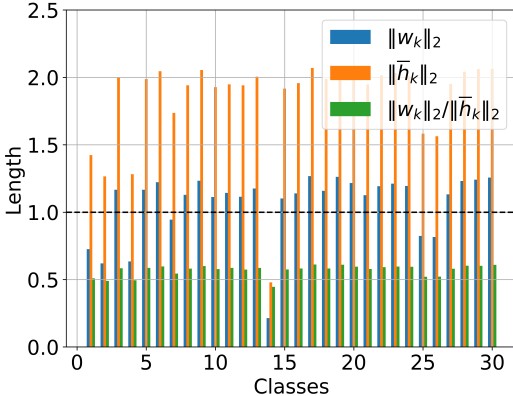

Figure B.2: Illustration of the length of the classifier weights and class-mean features in unconstrained feature model (UFM) with weight decay (WD) form in Figure B.1c. The ratios between the lengths of classifier weights and class-mean features vary across different classes.

## B.3 Additional results on prevalence of $\mathcal{GNC}$

We provide additional evidence on the occurrence of the $\mathcal{GNC}$ phenomenon in practical multi-class classification problems. Towards that, we train ResNet18, DenseNet121, and ResNeXt50 network on the CIFAR100, Tiny-ImageNet and BUPT-CBFace-50 datasets using CE loss. To illustrate the case where the feature dimension is smaller than the number of classes, we insert another linear layer before the last-layer classifier and set the dimensions of the features as $d = 10$ for CIFAR100 and Tiny-ImageNet, and $d = 512$ for BUPT-CBFace-50.

The results are reported in Figure B.3. It can be seen that in all the cases the $\mathcal{GNC}_1$, $\mathcal{GNC}_2$ and $\mathcal{GNC}_3$ measures converge mostly monotonically as a function of the training epochs towards the

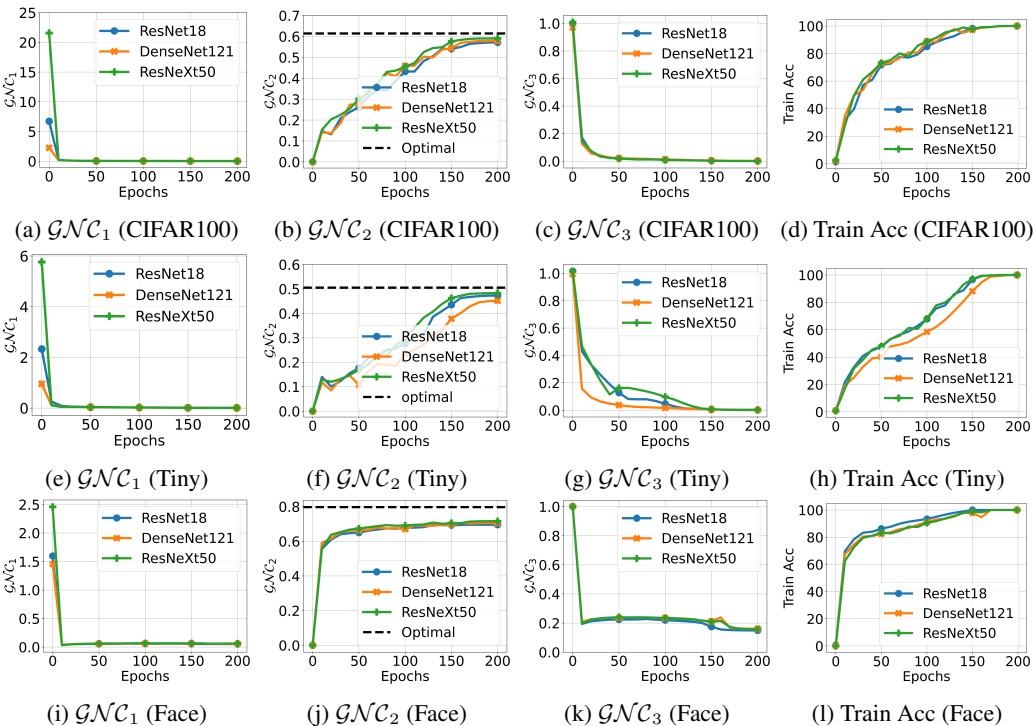

Figure B.3: **Illustration of $\mathcal{GNC}$ and train accuracy across different network architectures on CIFAR100(top), Tiny-ImageNet(middle) and BUPT-CBFace-50(bottom) datasets.** We train the networks on CIFAR100 with $d = 10$, $K = 100$, Tiny-ImageNet with $d = 10$, $K = 200$ and BUPT-CBFace-50 with $d = 512$, $K = 10000$.

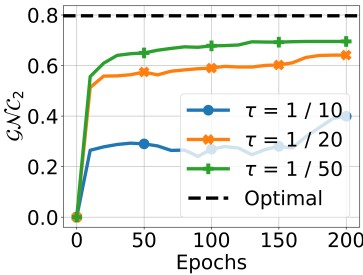

Figure B.4: **Illustration of $\mathcal{GNC}_2$ on BUPT-CBFace-50 dataset across varying and temperatures.** We train the networks on BUPT-CBFace-50 with $d = 512$, $K = 10,000$.

values predicted by $\mathcal{GNC}$(i.e., 0 for $\mathcal{GNC}_1$ and $\mathcal{GNC}_3$ and the objective of Softmax Code for $\mathcal{GNC}_2$, see Section 2.2).

**Effect of temperature $\tau$.** The results on BUPT-CBFace-50 reported in Figure B.3 uses a temperature $\tau = 0.02$. To examine the effect of $\tau$, we conduct experiments with varying $\tau$ and report the $\mathcal{GNC}_2$ in Figure B.4. It can be seen that the $\mathcal{GNC}_2$ measure at convergence monotonically increases as $\tau$ decreases.

**Implementation detail.** We choose the temperature $\tau = 0.1$ for CIFAR100 and Tiny-ImageNet and $\tau = 0.02$ for the BUPT-CBFace-50 dataset. The CIFAR100 dataset consists of $60,000$ $32 \times 32$ color images in 100 classes and Tiny-ImageNet contains $100,000$ $64 \times 64$ color images in 200 classes. The BUPT-CBFace-50 dataset consists of $500,000$ images in $10,000$ classes and all images are resized to the size of $50 \times 50$. To adapt to the smaller size images, we modify these architectures by changing the first convolutional layer to have a kernel size of 3, a stride of 1, and a padding

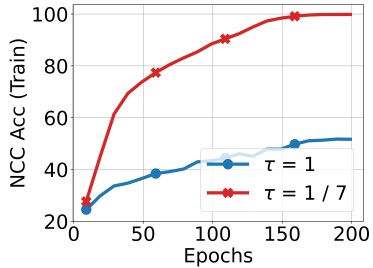

Figure B.5: **Illustration of NCC training accuracy with different temperatures.**

size of $1$. For our data augmentation strategy, we employ the random crop with a size of $32$ and padding of $4$, random horizontal flip with a probability of $0.5$, and random rotation with degrees of $15$ to increase the diversity of our training data. Then we normalize the images (channel-wise) using their mean and standard deviation. For optimization, we utilized SGD with a momentum of $0.9$ and an initial learning rate of $0.1$, which decayed according to the CosineAnnealing over a span of $200$ epochs. The optimal margin (represented by the dash line) in the second from left column is obtained through numerical optimization of the Softmax Codes problem.

### B.4 THE NEAREST CENTROID CLASSIFIER

As the learned features exhibit within-class variability collapse and are maximally distant between classes, the classifier also converges to the nearest centroid classifier (a.k.a nearest class-center classifier (NCC), where each sample is classified with the nearest class-mean features), which is termed as $\mathcal{NC}_4$ in Papyan et al. (2020) and exploited in (Galanti et al., 2022b;a; Rangamani et al., 2023) for studying $\mathcal{NC}$. To evaluate the convergence in terms of NCC accuracy, we use the same setup as in Figure 2, i.e., train a ResNet18 network on the CIFAR100 dataset with CE loss using different temperatures. We then classify the features by the NCC. The result is presented in Figure B.5. We can observe that with a relatively small temperature $\tau$, the NCC accuracy converges to 100% and hence the classifier also converges to a NCC.

### B.5 ADDITIONAL RESULTS ON THE EFFECT OF ASSIGNMENT PROBLEM

We provide additional evidence on the effect of "class assignment" problem in practical multi-class classification problems. To illustrate this, we train a ResNet18 network with d = 2 on six classes {Automobile, Cat, Deer, Dog, Horse, Truck} from CIFAR10 dataset that are selected due to their clear semantic similarity and discrepancy. Consequently, according to Theorem 3.3, there are fifteen distinct class assignments up to permutation.

When doing standard training, the classifier consistently converges to the case where Cat-Dog, Automobile-Truck and Deer-Horse pairs are closer together across 5 different trials; Figure B.6a shows the learned features (dots) and classifier weights (arrows) in one of such trials. This demonstrates the implicit algorithmic regularization in training DNNs, which naturally attracts (semantically) similar classes and separates dissimilar ones.

We also conduct experiments with the classifier fixed to be three of the fifteen arrangements, and present the results in Figures B.6b and B.6d. Among them, we observe that the case where Cat-Dog, Automobile-Truck and Deer-Horse pairs are far apart achieves a testing accuracy of $86.37\%$, which is lower than the other two cases with testing accuracies of $91.60\%$ and $91.95\%$. This demonstrates the important role of class assignment to the generalization of DNNs, and that the implicit bias of the learned classifier is benign, i.e., leads to a more generalizable solutions. Moreover, compared with the case of four classes in Figure 4, the discrepancy of test accuracy between different assignments increases from $2.18\%$ to $5.58\%$. This suggests that as the number of classes grows, the importance of appropriate class assignments becomes increasingly paramount.

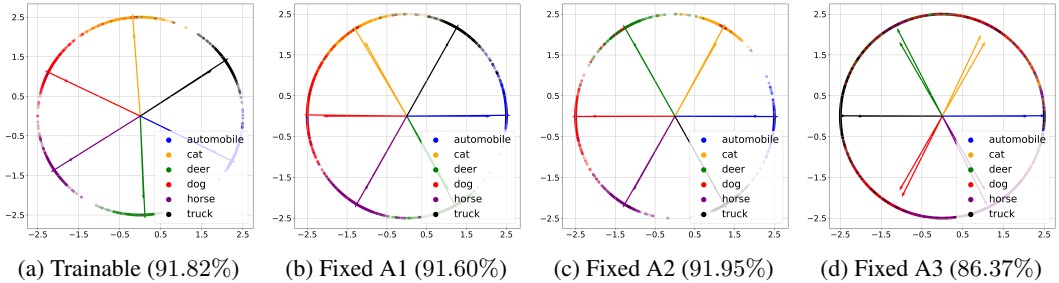

(a) Trainable (91.82%)     (b) Fixed A1 (91.60%)     (c) Fixed A2 (91.95%)     (d) Fixed A3 (86.37%)

Figure B.6: **Assignment of classes to classifier weights** for a ResNet18 with 2-dimensional feature space trained on the 6 classes {Automobile, Cat, Deer, Dog, Horse, Truck} from CIFAR10. *(a)* Learned classifier. *(b-d)* Classifiers fixed to be three different assignments. Test accuracy is reported in the bracket.

## C  THEORETICAL PROOFS

### C.1  PROOF OF LEMMA 3.1

We rewrite Lemma 3.1 below for convenience.

**Lemma C.1** (Convergence to the "HardMax" problem). *For any positive integers $K$ and $n$, we have*

$$\limsup_{\tau \to 0} \left( \arg\min \frac{1}{nK} \sum_{k=1}^{K} \sum_{i=1}^{n} \mathcal{L}_{CE} \left( \boldsymbol{W}^\top \boldsymbol{h}_{k,i}, \boldsymbol{y}_k, \tau \right) \right) \subseteq \arg\min \mathcal{L}_{HardMax}(\boldsymbol{W}, \boldsymbol{H}). \quad (\text{C.1})$$

*In above, all* $\arg\min$ *are taken over* $\boldsymbol{W} \in \mathcal{O}\mathrm{B}(d, K), \boldsymbol{H} \in \mathcal{O}\mathrm{B}(d, nK)$.

*Proof.* Our proof uses the fundamental theorem of $\Gamma$-convergence Braides (2006) (a.k.a. epi-convergence Rockafellar & Wets (2009)). Denote

$$\mathcal{L}_0(\boldsymbol{W}, \boldsymbol{H}, \tau) \doteq \tau \cdot \log \sum_{i=1}^{n} \sum_{k=1}^{K} \mathcal{L}_{\mathrm{CE}}(\boldsymbol{W}^\top \boldsymbol{h}_{k,i}, \boldsymbol{y}_k, \tau), \quad (\text{C.2})$$

and let

$$\mathcal{WH}^- \doteq \{(\boldsymbol{W}, \boldsymbol{H}) \in \mathcal{O}\mathrm{B}(d, K) \times \mathcal{O}\mathrm{B}(d, K) : (\boldsymbol{w}_{k'} - \boldsymbol{w}_k)^\top \boldsymbol{h}_{k,i} \le 0, \forall i \in [n], k \in [K], k' \in [K] \backslash k\}. \quad (\text{C.3})$$

By Lemma C.2, the function $\mathcal{L}_0(\boldsymbol{W}, \boldsymbol{H}, \tau)$ converges uniformly to $\mathcal{L}_{\mathrm{HardMax}}(\boldsymbol{W}, \boldsymbol{H})$ on $\mathcal{WH}^-$ as $\epsilon \to 0$. Combining it with (Rockafellar & Wets, 2009, Proposition 7.15), we have $\mathcal{L}_0(\boldsymbol{W}, \boldsymbol{H}, \tau)$ $\Gamma$-converges to $\mathcal{L}_{\mathrm{HardMax}}(\boldsymbol{W}, \boldsymbol{H})$ on $\mathcal{WH}^-$ as well. By applying (Braides, 2006, Theorem 2.10), we have

$$\limsup_{\tau \to 0} \arg\min_{(\boldsymbol{W}, \boldsymbol{H}) \in \mathcal{WH}^-} \mathcal{L}_0(\boldsymbol{W}, \boldsymbol{H}, \tau) \subseteq \arg\min_{(\boldsymbol{W}, \boldsymbol{H}) \in \mathcal{WH}^-} \mathcal{L}_{\mathrm{HardMax}}(\boldsymbol{W}, \boldsymbol{H}). \quad (\text{C.4})$$

Note that in above, $\arg\min$ are taken over $\mathcal{WH}^-$ which is a strict subset of $\boldsymbol{W} \in \mathcal{O}\mathrm{B}(d, K), \boldsymbol{H} \in \mathcal{O}\mathrm{B}(d, nK)$. However, by Lemma C.3, we know that

$$\arg\min_{(\boldsymbol{W}, \boldsymbol{H}) \in \mathcal{WH}^-} \mathcal{L}_0(\boldsymbol{W}, \boldsymbol{H}, \tau) = \arg\min_{(\boldsymbol{W}, \boldsymbol{H}) \in \mathcal{O}\mathrm{B}(d, K) \times \mathcal{O}\mathrm{B}(d, K)} \mathcal{L}_0(\boldsymbol{W}, \boldsymbol{H}, \tau), \quad (\text{C.5})$$

which holds for all $\tau$ sufficiently small. Hence, we have

$$\limsup_{\tau \to 0} \arg\min_{(\boldsymbol{W}, \boldsymbol{H}) \in \mathcal{O}\mathrm{B}(d, K) \times \mathcal{O}\mathrm{B}(d, K)} \mathcal{L}_0(\boldsymbol{W}, \boldsymbol{H}, \tau) \subseteq \arg\min_{(\boldsymbol{W}, \boldsymbol{H}) \in \mathcal{O}\mathrm{B}(d, K) \times \mathcal{O}\mathrm{B}(d, K)} \mathcal{L}_{\mathrm{HardMax}}(\boldsymbol{W}, \boldsymbol{H}),$$
$$(\text{C.6})$$

which concludes the proof. □

**Lemma C.2.** $\mathcal{L}_0(\boldsymbol{W}, \boldsymbol{H}, \tau)$ *converges uniformly to* $\mathcal{L}_{HardMax}(\boldsymbol{W}, \boldsymbol{H})$ *in the domain* $(\boldsymbol{W}, \boldsymbol{H}) \in \mathcal{WH}^-$ *as* $\tau \to 0$.

*Proof.* Recall from the definition of the CE loss in (1) that

$$\mathcal{L}_0(\boldsymbol{W}, \boldsymbol{H}, \tau) \doteq \tau \cdot \log \sum_{i=1}^{n} \sum_{k=1}^{K} \mathcal{L}_{\mathrm{CE}}(\boldsymbol{W}^\top \boldsymbol{h}_{k,i}, \boldsymbol{y}_k, \tau)$$

$$= \tau \log \sum_{i=1}^{n} \sum_{k=1}^{K} \log \left(1 + \sum_{k' \in [K] \setminus k} \exp \left((\boldsymbol{w}_{k'} - \boldsymbol{w}_k)^\top \boldsymbol{h}_{k,i}/\tau\right)\right) \quad \text{(C.7)}$$

Denote $\alpha_{i,k,k'} = (\boldsymbol{w}_{k'} - \boldsymbol{w}_k)^\top \boldsymbol{h}_{k,i}, \forall i \in [n], k \in [K], k' \in [K] \setminus k$ for convenience. Fix any $i \in [n]$ and $k \in [K]$, by the property that $\frac{x}{1+x} \le \log(1+x) \le x$ for all $x > -1$, we have

$$\frac{\sum_{k' \in [K] \setminus k} \exp \left(\frac{\alpha_{i,k,k'}}{\tau}\right)}{1 + \sum_{k' \in [K] \setminus k} \exp \left(\frac{\alpha_{i,k,k'}}{\tau}\right)} \le \log \left(1 + \sum_{k' \in [K] \setminus k} \exp \left(\frac{\alpha_{i,k,k'}}{\tau}\right)\right) \le \sum_{k' \in [K] \setminus k} \exp \left(\frac{\alpha_{i,k,k'}}{\tau}\right).$$
$$\text{(C.8)}$$

Note that in the domain $(\boldsymbol{W}, \boldsymbol{H}) \in \mathcal{WH}^-$ we have $\alpha_{i,k,k'} \le 0$ for all $i, k, k' \ne k$. It follows trivially from monotonicity of exponential function that $\sum_{k' \in [K] \setminus k} \exp \left(\frac{\alpha_{i,k,k'}}{\tau}\right) < K - 1$ holds for all $i, k$. Hence, continuing from the inequality above we have

$$\frac{\sum_{k' \in [K] \setminus k} \exp \left(\frac{\alpha_{i,k,k'}}{\tau}\right)}{K} \le \log \left(1 + \sum_{k' \in [K] \setminus k} \exp \left(\frac{\alpha_{i,k,k'}}{\tau}\right)\right) \le \sum_{k' \in [K] \setminus k} \exp \left(\frac{\alpha_{i,k,k'}}{\tau}\right). \quad \text{(C.9)}$$

Summing over $i, k$ we obtain

$$\sum_{i,k} \frac{\sum_{k' \in [K] \setminus k} \exp \left(\frac{\alpha_{i,k,k'}}{\tau}\right)}{K} \le \sum_{i,k} \log \left(1 + \sum_{k' \in [K] \setminus k} \exp \left(\frac{\alpha_{i,k,k'}}{\tau}\right)\right) \le \sum_{i,k} \sum_{k' \in [K] \setminus k} \exp \left(\frac{\alpha_{i,k,k'}}{\tau}\right),$$
$$\text{(C.10)}$$

Using the property of max function we further obtain

$$\frac{\max_{i,k,k' \in [K] \setminus k} \exp \left(\frac{\alpha_{i,k,k'}}{\tau}\right)}{K} \le \sum_{i,k} \log \left(1 + \sum_{k' \in [K] \setminus k} \exp \left(\frac{\alpha_{i,k,k'}}{\tau}\right)\right)$$

$$\le n \cdot K \cdot (K-1) \cdot \max_{i,k,k' \in [K] \setminus k} \exp \left(\frac{\alpha_{i,k,k'}}{\tau}\right). \quad \text{(C.11)}$$

Taking logarithmic on both sides and multiplying all terms by $\tau$ we get

$$\max_{i,k,k' \in [K] \setminus k} \alpha_{i,k,k'} - \tau \log K \le \mathcal{L}_0(\boldsymbol{W}, \boldsymbol{H}, \tau) \le \tau \log(n \cdot K \cdot (K-1)) + \max_{i,k,k' \in [K] \setminus k} \alpha_{i,k,k'}. \quad \text{(C.12)}$$

Noting that $\max_{i,k,k' \in [K] \setminus k} \alpha_{i,k,k'} = \mathcal{L}_{\mathrm{HardMax}}(\boldsymbol{W}, \boldsymbol{H})$, we have

$$\mathcal{L}_{\mathrm{HardMax}}(\boldsymbol{W}, \boldsymbol{H}) - \tau \log K \le \mathcal{L}_0(\boldsymbol{W}, \boldsymbol{H}, \tau) \le \tau \log(n \cdot K \cdot (K-1)) + \mathcal{L}_{\mathrm{HardMax}}(\boldsymbol{W}, \boldsymbol{H}). \quad \text{(C.13)}$$

Hence, for any $\epsilon > 0$, by taking $\tau_0 = \frac{\epsilon}{\max\{\log K, \log(n \cdot K \cdot (K-1))\}}$, we have that for any $(\boldsymbol{W}, \boldsymbol{H}) \in \mathcal{WH}^-$, we have

$$|\mathcal{L}_0(\boldsymbol{W}, \boldsymbol{H}, \tau) - \mathcal{L}_{\mathrm{HardMax}}(\boldsymbol{W}, \boldsymbol{H})| \le \tau \max\{\log K, \log(n \cdot K \cdot (K-1))\} < \epsilon, \quad \text{(C.14)}$$

for any $\tau < \tau_0$. That is, $\mathcal{L}_0(\boldsymbol{W}, \boldsymbol{H}, \tau)$ converges uniformly to $\mathcal{L}_{\mathrm{HardMax}}(\boldsymbol{W}, \boldsymbol{H})$.

$\square$

**Lemma C.3.** *Given any $n, K$ there exists a constant $\tau_0$ such that for any $\tau < \tau_0$ we have*

$$\underset{(\boldsymbol{W}, \boldsymbol{H}) \in \mathcal{OB}(d,K) \times \mathcal{OB}(d,K)}{\arg \min} \mathcal{L}_0(\boldsymbol{W}, \boldsymbol{H}, \tau) \in \mathcal{WH}^-. \quad \text{(C.15)}$$

*Proof.* Note that for any $(\boldsymbol{W}, \boldsymbol{H}) \notin \mathcal{O}\mathrm{B}(d, K) \times \mathcal{O}\mathrm{B}(d, K)$, there exists $\bar{i} \in [n], \bar{k} \in [K]$, and $\bar{k}' \in [K] \setminus \bar{k}$ such that $(\boldsymbol{w}_{\bar{k}'} - \boldsymbol{w}_{\bar{k}})^\top \boldsymbol{h}_{\bar{k}, \bar{i}} \geq 0$. Hence, we have

$$\mathcal{L}_0(\boldsymbol{W}, \boldsymbol{H}, \tau) = \tau \log \sum_{i=1}^{n} \sum_{k=1}^{K} \log \left( 1 + \sum_{k' \in [K] \setminus k} \exp \left( (\boldsymbol{w}_{k'} - \boldsymbol{w}_k)^\top \boldsymbol{h}_{k,i} / \tau \right) \right)$$

$$\geq \tau \log \log(1 + \exp(\boldsymbol{w}_{\bar{k}'} - \boldsymbol{w}_{\bar{k}})^\top \boldsymbol{h}_{\bar{k}, \bar{i}} / \tau) \geq \tau \log \log(2), \quad \forall \tau > 0. \quad \text{(C.16)}$$

On the other hand, we show that one may construct a $(\boldsymbol{W}^*, \boldsymbol{H}^*) \in \mathcal{W}\mathcal{H}^-$ such that $\mathcal{L}_0(\boldsymbol{W}^*, \boldsymbol{H}^*, \tau) < \tau \log \log(2)$ for any small enough $\tau$. Towards that, we take $\boldsymbol{W}^*$ to be any matrix in $\mathcal{O}\mathrm{B}(d, K)$ with distinct columns. Denote $M = \max_{k \neq k'} \langle \boldsymbol{w}_k^*, \boldsymbol{w}_{k'}^* \rangle$ the inner product of the closest pair of columns from $\boldsymbol{W}^*$, which by construction has value $M < 1$. Take $\boldsymbol{h}_{k,i} = \boldsymbol{w}_k$ for all $k \in [K], i \in [n]$. We have $(\boldsymbol{w}_{k'}^* - \boldsymbol{w}_k^*)^\top \boldsymbol{h}_{k,i}^* \leq -(1 - M)$, for all $i \in [n], k \in [K]$, and $k' \in [K] \setminus k$. Plugging this into the definition of $\mathcal{L}_0(\boldsymbol{W}, \boldsymbol{H}, \tau)$ we have

$$\mathcal{L}_0(\boldsymbol{W}^*, \boldsymbol{H}^*, \tau) = \tau \log \sum_{i=1}^{n} \sum_{k=1}^{K} \log \left( 1 + \sum_{k' \in [K] \setminus k} \exp \left( (\boldsymbol{w}_{k'}^* - \boldsymbol{w}_k^*)^\top \boldsymbol{h}_{k,i}^* / \tau \right) \right)$$

$$\leq \tau \log \left( nK \log(1 + (K-1) \exp\left(-\frac{1-M}{\tau}\right)) \right) < \tau \log \log 2, \quad \forall \tau < \tau_0 \quad \text{(C.17)}$$

for some $\tau_0 > 0$ that depends only on $M$. In above, the last inequality holds because one can always find a $\tau_0 > 0$ such that $nK \log(1 + (K-1) \exp\left(-\frac{1-M}{\tau}\right)) < \log 2$ for $\tau < \tau_0$. This implies that any $(\boldsymbol{W}, \boldsymbol{H}) \notin \mathcal{O}\mathrm{B}(d, K) \times \mathcal{O}\mathrm{B}(d, K)$ is not a minimizer of $\mathcal{L}_0(\boldsymbol{W}, \boldsymbol{H}, \tau)$ for a sufficiently small $\tau$, which finishes the proof. $\qquad \square$

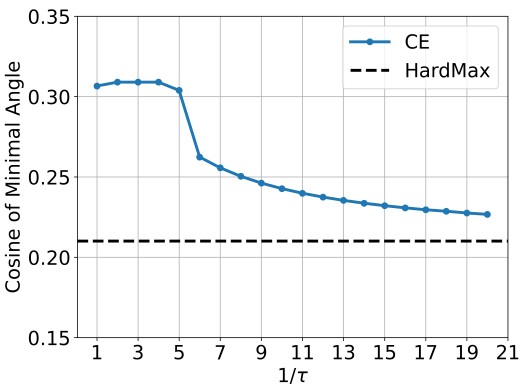

Figure C.1: Verifying Lemma 3.1 under $d = 3$ and $K = 7$.

As depicted in Figure C.1, we plot the cosine value of the minimal angle obtained from optimizing the CE loss (blue line) and the "HardMax" approach (black line) for different temperature parameters. The figure demonstrates that as the temperature parameter $\tau \to 0$, the blue line converges to the black line, thus validating our proof.

## C.2 AN IMPORTANT LEMMA

We present an important lemma which will be used to prove many of the subsequent results.

**Lemma C.4** (Optimal Features for Fixed Classifier). *For any $k \in [K]$, suppose $\boldsymbol{w}_k \notin \mathrm{conv}(\{\boldsymbol{w}_j\}_{j \in [K] \setminus k})$, then*

$$\min_{\boldsymbol{h} \in \mathbb{S}^{d-1}} \max_{k' \neq k} \langle \boldsymbol{w}_{k'} - \boldsymbol{w}_k, \boldsymbol{h} \rangle = -\mathrm{dist}(\boldsymbol{w}_k, \{\boldsymbol{w}_j\}_{j \in [K] \setminus k}). \quad \text{(C.18)}$$

*In addition, the optimal $\boldsymbol{h}$ is given by*

$$\boldsymbol{h} = \mathcal{P}_{\mathbb{S}^{d-1}} \left( \boldsymbol{w}_k - \mathcal{P}_{\{\boldsymbol{w}_j\}_{j \in [K] \setminus k}}(\boldsymbol{w}_k) \right) \quad \text{(C.19)}$$

where $\mathcal{P}_{\mathcal{W}}(\boldsymbol{v}) \doteq \arg\min_{\boldsymbol{w}\in\text{conv}(\mathcal{W})}\{\|\boldsymbol{v}-\boldsymbol{w}\|_2\}$ denotes the projection of $\boldsymbol{v}$ on $\text{conv}(\mathcal{W})$.

*Proof.* The proof follows from combining Lemma C.5 and Lemma C.6. □

**Lemma C.5.** *Suppose* $\boldsymbol{w}_k \notin \text{conv}(\{\boldsymbol{w}_j\}_{j\in[K]\setminus k})$. *Then*

$$\min_{\boldsymbol{h}\in\mathbb{S}^{d-1}} \max_{k'\neq k} \langle \boldsymbol{w}_{k'}-\boldsymbol{w}_k, \boldsymbol{h}\rangle \equiv \min_{\|\boldsymbol{h}\|_2\leq 1} \max_{k'\neq k} \langle \boldsymbol{w}_{k'}-\boldsymbol{w}_k, \boldsymbol{h}\rangle \qquad (\text{C.20})$$

*where* $\equiv$ *means the two problems are equivalent, i.e., have the same optimal solutions.*

*Proof of Lemma C.5.* By the separating hyperplane theorem (e.g. (Boyd & Vandenberghe, 2004, Example 2.20)), there exist a nonzero vector $\bar{\boldsymbol{h}}$ and $b \in \mathbb{R}$ such that $\max_{k'\neq k}\langle \boldsymbol{w}_{k'}, \bar{\boldsymbol{h}}\rangle < b$ and $\langle \boldsymbol{w}_k, \bar{\boldsymbol{h}}\rangle > b$, i.e., $\max_{k'\neq k}\langle \boldsymbol{w}_{k'}-\boldsymbol{w}_k, \bar{\boldsymbol{h}}\rangle < 0$. Let $\boldsymbol{h}^*$ be any optimal solution to the RHS of (C.20). Then, it holds that

$$\max_{k'\neq k} \langle \boldsymbol{w}_{k'}-\boldsymbol{w}_k, \boldsymbol{h}^*\rangle \leq \max_{k'\neq k} \left\langle \boldsymbol{w}_{k'}-\boldsymbol{w}_k, \frac{\bar{\boldsymbol{h}}}{\|\bar{\boldsymbol{h}}\|_2}\right\rangle < 0.$$

Hence, it must be the case that $\|\boldsymbol{h}^*\|_2 = 1$; otherwise, taking $\boldsymbol{h} = \boldsymbol{h}^*/\|\boldsymbol{h}^*\|_2$ gives lower objective for the RHS of (C.20), contradicting the optimality of $\boldsymbol{h}^*$. □

**Lemma C.6.** *Suppose* $\boldsymbol{w}_k \notin \text{conv}(\{\boldsymbol{w}_j\}_{j\in[K]\setminus k})$. *Consider the (primal) problem*

$$\min_{\|\boldsymbol{h}\|_2\leq 1} \max_{k'\neq k} \langle \boldsymbol{w}_{k'}-\boldsymbol{w}_k, \boldsymbol{h}\rangle. \qquad (\text{C.21})$$

*Its dual problem is given by*

$$\max_{\boldsymbol{v}\in\mathbb{R}^d} -\|\boldsymbol{w}_k - \sum_{k'\neq k} v_{k'}\boldsymbol{w}_{k'}\|_2 \ \ s.t. \ \sum_{k'\neq k} v_{k'} = 1, \ and \ v_{k'} \geq 0, \forall k'\neq k. \qquad (\text{C.22})$$

*with zero duality gap. Moreover, for any primal optimal solution* $\boldsymbol{h}^*$ *there is a dual optimal solution* $\boldsymbol{v}^*$ *and they satisfy*

$$\boldsymbol{h}^* = \frac{\boldsymbol{w}_k - \sum_{k'} v_{k'}^* \boldsymbol{w}_{k'}}{\|\boldsymbol{w}_k - \sum_{k'} v_{k'}^* \boldsymbol{w}_{k'}\|_2}. \qquad (\text{C.23})$$

*Proof of Lemma C.6.* We rewrite the primal problem as

$$\min_{\|\boldsymbol{h}\|_2\leq 1, \boldsymbol{p}\in\mathbb{R}^d} \max_{k'\neq k} p_{k'} \ \ \text{s.t.} \ p_{k'} = \langle \boldsymbol{w}_{k'}-\boldsymbol{w}_k, \boldsymbol{h}\rangle. \qquad (\text{C.24})$$

Introducing the dual variable $\boldsymbol{v} \in \mathbb{R}^d$, the Lagragian function is

$$\mathcal{L}(\boldsymbol{h},\boldsymbol{p},\boldsymbol{v}) = \max_{k'\neq k} p_{k'} - \sum_{k'\neq k} v_{k'}(p_{k'} - \langle \boldsymbol{w}_{k'}-\boldsymbol{w}_k, \boldsymbol{h}\rangle), \ \ \|\boldsymbol{h}\|_2 \leq 1. \qquad (\text{C.25})$$

We now derive the dual problem, defined as

$$\max_{\boldsymbol{v}} \min_{\|\boldsymbol{h}\|_2\leq 1, \boldsymbol{p}\in\mathbb{R}^d} \max_{k'\neq k} \mathcal{L}(\boldsymbol{h},\boldsymbol{p},\boldsymbol{v}) = \max_{\boldsymbol{v}} \left( \min_{\boldsymbol{p}} \left( \max_{k'\neq k} p_{k'} - \sum_{k'\neq k} v_{k'} p_{k'} \right) + \min_{\|\boldsymbol{h}\|_2\leq 1} \left\langle \sum_{k'\neq k} v_{k'}\boldsymbol{w}_{k'} - \boldsymbol{w}_k, \boldsymbol{h} \right\rangle \right)$$

$$= \max_{\boldsymbol{v}} \min_{\|\boldsymbol{h}\|_2\leq 1} \left\langle \sum_{k'\neq k} v_{k'}\boldsymbol{w}_{k'} - \boldsymbol{w}_k, \boldsymbol{h} \right\rangle \ \ \text{s.t.} \ \sum_{k'\neq k} v_{k'} = 1, \ v_{k'} \geq 0 \ \forall k'\neq k$$

$$= \max_{\boldsymbol{v}} -\|\boldsymbol{w}_k - \sum_{k'\neq k} v_{k'}\boldsymbol{w}_{k'}\|_2 \ \ \text{s.t.} \ \sum_{k'\neq k} v_{k'} = 1, \ v_{k'} \geq 0 \ \forall k'\neq k.$$

$$(\text{C.26})$$

In above, the second equality follows from the fact that the conjugate (see e.g. Boyd & Vandenberghe (2004)) of the max function is the indicator function of the probability simplex. The third equality uses the assumption that $\boldsymbol{w}_k \notin \text{conv}(\{\boldsymbol{w}_j\}_{j\in[K]\setminus k})$, which implies that $\sum_{k'\neq k} v_{k'}\boldsymbol{w}_{k'} - \boldsymbol{w}_k \neq 0$ under the simplex constraint of $\boldsymbol{v}$, hence the optimal $\boldsymbol{h}$ to the optimization in the second line can be easily obtained as $\boldsymbol{h} = \frac{\boldsymbol{w}_k - \sum_{k'} v_{k'}\boldsymbol{w}_{k'}}{\|\boldsymbol{w}_k - \sum_{k'} v_{k'}\boldsymbol{w}_{k'}\|_2}$.

Finally, the rest of the claims hold as the primal problem is convex with the Slater's condition satisfied. □

## C.3 PROOF OF THEOREM 3.2

Here we prove the following result which is a stronger version of Theorem 3.2.

**Theorem C.7.** *Let $(\boldsymbol{W}^\star, \boldsymbol{H}^\star)$ be an optimal solution to (5). Then, it holds that $\boldsymbol{W}^\star$ is a Softmax Code, i.e.,*

$$\boldsymbol{W}^\star \in \underset{\boldsymbol{W} \in \mathcal{OB}(d,K)}{\arg\max} \; \rho_{\text{one-vs-rest}}(\boldsymbol{W}). \tag{C.27}$$

*Conversely, let $\boldsymbol{W}^{SC}$ be any Softmax Code. Then, there exists a $\boldsymbol{H}^{SC}$ such that $(\boldsymbol{W}^{SC}, \boldsymbol{H}^{SC})$ is an optimal solution to (5).*

*Proof.* The proof is divided into two parts.

**Any optimal solution to (5) is a Softmax Code.** Our proof is based on providing a lower bound on the objective $\mathcal{L}_{\text{HardMax}}(\boldsymbol{W}, \boldsymbol{H})$. We distinguish two cases in deriving the lower bound.

- $\boldsymbol{W}$ has distinct columns. In this case we use the following bound:

$$\mathcal{L}_{\text{HardMax}}(\boldsymbol{W}, \boldsymbol{H}) = \max_{k \in [K]} \max_{i \in [n]} \max_{k' \neq k} \langle \boldsymbol{w}_{k'} - \boldsymbol{w}_k, \boldsymbol{h}_{k,i} \rangle$$
$$\geq \max_{k \in [K]} \max_{i \in [n]} \min_{\bar{\boldsymbol{h}}_{k,i} \in \mathbb{S}^{d-1}} \max_{k' \neq k} \langle \boldsymbol{w}_{k'} - \boldsymbol{w}_k, \bar{\boldsymbol{h}}_{k,i} \rangle = - \min_{k \in [K]} \text{dist}(\boldsymbol{w}_k, \{\boldsymbol{w}_j\}_{j \in [K] \setminus k}).$$
$$\tag{C.28}$$

  In above, the first equality follows directly from definition of the HardMax function. The inequality follows trivially from the property of the $\min$ operator. The last equality follows from Lemma C.4, which requires that $\boldsymbol{W}$ has distinct columns. Continuing on the rightmost term in (C.28), we have

$$- \min_{k \in [K]} \text{dist}(\boldsymbol{w}_k, \{\boldsymbol{w}_j\}_{j \in [K] \setminus k}) = -\rho_{\text{one-vs-rest}}(\boldsymbol{W}) \geq -\rho_{\text{one-vs-rest}}(\boldsymbol{W}^{SC}), \tag{C.29}$$

  where $\boldsymbol{W}^{SC}$ is any Softmax Code. In above, the equality follows from the definition of the operator $\rho_{\text{one-vs-rest}}()$, and the inequality follows from the definition of the Softmax Code. In particular, by defining $\widehat{\boldsymbol{W}} = \boldsymbol{W}^{SC}$ and $\widehat{\boldsymbol{H}}$ as

$$\widehat{\boldsymbol{h}}_{k,i} = \frac{\widehat{\boldsymbol{w}}_k - \text{proj}(\widehat{\boldsymbol{w}}_k, \{\widehat{\boldsymbol{w}}_j\}_{j \in [K] \setminus k})}{\|\widehat{\boldsymbol{w}}_k - \text{proj}(\widehat{\boldsymbol{w}}_k, \{\widehat{\boldsymbol{w}}_j\}_{j \in [K] \setminus k})\|_2}, \tag{C.30}$$

  all inequalities in (C.28) and (C.29) holds with equality by taking $\boldsymbol{W} = \widehat{\boldsymbol{W}}$ and $\boldsymbol{H} = \widehat{\boldsymbol{H}}$, at which we have that $\mathcal{L}_{\text{HardMax}}(\widehat{\boldsymbol{W}}, \widehat{\boldsymbol{H}}) = -\rho_{\text{one-vs-rest}}(\boldsymbol{W}^{SC})$.

- $\boldsymbol{W}$ does not have distinct columns. Hence, there exists $k_1$, $k_2$ such that $k_1 \neq k_2$ but $\boldsymbol{w}_{k_1} = \boldsymbol{w}_{k_2}$. We have

$$\mathcal{L}_{\text{HardMax}}(\boldsymbol{W}, \boldsymbol{H}) = \max_{k \in [K]} \max_{i \in [n]} \max_{k' \neq k} \langle \boldsymbol{w}_{k'} - \boldsymbol{w}_k, \boldsymbol{h}_{k,i} \rangle \geq \max_{i \in [n]} \langle \boldsymbol{w}_{k_2} - \boldsymbol{w}_{k_1}, \boldsymbol{h}_{k_1,i} \rangle = 0. \tag{C.31}$$

Combining the above two cases, and by noting that $-\rho_{\text{one-vs-rest}}(\boldsymbol{W}^{SC}) < 0$, we have that $(\widehat{\boldsymbol{W}}, \widehat{\boldsymbol{H}})$ is an optimal solution to the HardMax problem in (5). Moreover, since $(\boldsymbol{W}^*, \boldsymbol{H}^*)$ is an optimal solution to the HardMax problem, it must attain the lower bound, i.e.,

$$\mathcal{L}_{\text{HardMax}}(\boldsymbol{W}^*, \boldsymbol{H}^*) = -\rho_{\text{one-vs-rest}}(\boldsymbol{W}^{SC}). \tag{C.32}$$

Hence, $\boldsymbol{W}^*$ has to attain the equality in (C.29). By definition, this means that $\boldsymbol{W}^*$ is a Softmax Code, which concludes the proof of this part.

**From any Softmax Code we can construct an optimal solution to (5).** Let

$$\boldsymbol{W}^{SC} \in \underset{\boldsymbol{W} \in \mathcal{OB}(d,K)}{\arg\max} \; \rho_{\text{one-vs-rest}}(\boldsymbol{W}) \tag{C.33}$$

be any Softmax Code. Moreover, define $\boldsymbol{H}^{\text{SC}}$ to be such that

$$\boldsymbol{h}_{k,i}^{\text{SC}} = \underset{\boldsymbol{h}_k \in \mathbb{S}^{d-1}}{\arg\min}\, \underset{k' \neq k}{\max}\langle \boldsymbol{w}_{k'}^{\text{SC}} - \boldsymbol{w}_k^{\text{SC}}, \boldsymbol{h}_k \rangle, \forall k \in [K], \forall i \in [n]. \tag{C.34}$$

Note that the following result holds which will be used in the subsequent proof:

$$\begin{aligned}
\boldsymbol{W}^{\text{SC}} &\in \underset{\boldsymbol{W} \in \mathcal{O}\text{B}(d,K)}{\arg\max}\, \underset{k}{\min}\, \text{dist}\Big(\boldsymbol{w}_k, \{\boldsymbol{w}_j\}_{j \in [K]\setminus k}\Big) && \text{(Definition of Softmax Code)} \\
&\in \underset{\boldsymbol{W} \in \mathcal{O}\text{B}(d,K)}{\arg\min}\, \underset{k}{\max}\, \underset{\boldsymbol{h}_k \in \mathbb{S}^{d-1}}{\min}\, \underset{k' \neq k}{\max}\langle \boldsymbol{w}_{k'} - \boldsymbol{w}_k, \boldsymbol{h}_k \rangle. && \text{(Lemma C.4)}
\end{aligned} \tag{C.35}$$

For any $(\widehat{\boldsymbol{W}}, \widehat{\boldsymbol{H}})$, we have

$$\begin{aligned}
\mathcal{L}_{\text{HardMax}}(\widehat{\boldsymbol{W}}, \widehat{\boldsymbol{H}}) &= \underset{k \in [K]}{\max}\, \underset{i \in [n]}{\max}\, \underset{k' \neq k}{\max}\langle \widehat{\boldsymbol{w}}_{k'} - \widehat{\boldsymbol{w}}_k, \widehat{\boldsymbol{h}}_{k,i} \rangle && \text{(Definition of } \mathcal{L}_{\text{HardMax}}) \\
&\geq \underset{k \in [K]}{\max}\, \underset{\boldsymbol{h}_k \in \mathbb{S}^{d-1}}{\min}\, \underset{k' \neq k}{\max}\langle \widehat{\boldsymbol{w}}_{k'} - \widehat{\boldsymbol{w}}_k, \boldsymbol{h}_k \rangle \\
&\geq \underset{k \in [K]}{\max}\, \underset{\boldsymbol{h}_k \in \mathbb{S}^{d-1}}{\min}\, \underset{k' \neq k}{\max}\langle \boldsymbol{w}_{k'}^{\text{SC}} - \boldsymbol{w}_k^{\text{SC}}, \boldsymbol{h}_k \rangle && \text{(Eq. (C.35))} \\
&= \underset{k \in [K]}{\max}\, \underset{i \in [n]}{\max}\, \underset{k' \neq k}{\max}\langle \boldsymbol{w}_{k'}^{\text{SC}} - \boldsymbol{w}_k^{\text{SC}}, \boldsymbol{h}_{k,i}^{\text{SC}} \rangle && \text{(Eq. (C.34))} \\
&= \mathcal{L}_{\text{HardMax}}(\boldsymbol{W}^{\text{SC}}, \boldsymbol{H}^{\text{SC}}). && \text{(Definition of } \mathcal{L}_{\text{HardMax}})
\end{aligned} \tag{C.36}$$

This implies that $(\boldsymbol{W}^{\text{SC}}, \boldsymbol{H}^{\text{SC}})$ is an optimal solution to the HardMax problem in (5), which concludes the proof of this part.

$\square$

## C.4 PROOF OF THEOREM 3.3

**Theorem C.8.** *For any positive integers $K$ and $d$, let $\boldsymbol{W}^\star \in \mathcal{O}\text{B}(d,K)$ be a Softmax Code. Then,*

- *$d = 2$: $\{\boldsymbol{w}_k^\star\}$ is uniformly distributed on the unit circle, i.e., $\{\boldsymbol{w}_k^\star\} = \{\big(\cos(\frac{2\pi k}{K}+\alpha), \sin(\frac{2\pi k}{K}+\alpha)\big)\}$ for some $\alpha$;*

- *$K \leq d+1$: $\{\boldsymbol{w}_k^\star\}$ forms a simplex ETF, i.e., $\boldsymbol{W}^\star = \sqrt{\frac{K}{K-1}}\boldsymbol{P}(\boldsymbol{I}_K - \frac{1}{K}\boldsymbol{1}_K\boldsymbol{1}_K^\top)$ for some orthonomal $\boldsymbol{P} \in \mathbb{R}^{d \times K}$;*

- *$d+1 < K \leq 2d$: $\min_k \text{dist}(\boldsymbol{w}_k^\star, \{\boldsymbol{w}_j^\star\}_{j \in [K]\setminus k}) = 1$ which can be achieved when $\{\boldsymbol{w}_k^\star\}$ are a subset of vertices of a cross-polytope;*

- *$K \to \infty$: $\{\boldsymbol{w}_k^\star\}$ are uniformly distributed on the unite sphere $\mathbb{S}^{d-1}$;*

*Proof.* We prove the results case by case as follows.

- **$d = 2$: $\{w_k^\star\}$ are uniformly distributed on the unite sphere of $\mathbb{S}^1$.**

    Denote $\boldsymbol{w}_k^\star = [\cos \alpha_k^\star, \sin \alpha_k^\star], \forall k \in [K]$. Without loss of generality we may assume that $0 < \alpha_1^\star < \alpha_2^\star < \ldots < \alpha_K^\star \leq 2\pi$. Define

    $$\theta_k^\star = \begin{cases} \alpha_{k+1}^\star - \alpha_k^\star, & \forall k \in [K-1] \\ \alpha_1^\star - \alpha_K^\star + 2\pi, & k = K. \end{cases} \tag{C.37}$$

    For convenience, we also define $\alpha_{K+1}^\star \doteq \alpha_1^\star$ and $\theta_{K+1}^\star \doteq \theta_1^\star$.

    Geometrically, $\theta_k^\star$ is the angular distance between $\alpha_k^\star$ and $\alpha_{k+1}^*$. Moreover, by summing up all terms in (C.37) over $k \in [K]$ we have

    $$\sum_{k \in [K]} \theta_k^\star = 2\pi. \tag{C.38}$$

    To prove the theorem we only need to show that $\theta_k^\star = \frac{2\pi}{K}$ for all $k \in [K]$, which implies that $\{\boldsymbol{w}_k^\star\}$ are uniformly distributed on the unit circle.

We start by noting that the following result holds:

$$\theta_k^\star + \theta_{k+1}^\star \geq 2 \times \frac{2\pi}{K}, \quad \forall k \in [K]. \tag{C.39}$$

To see why, let $\{\widehat{\boldsymbol{w}}_k = [\cos\widehat{\alpha}_k, \sin\widehat{\alpha}_k]\}_{k\in[K]}$ with $\widehat{\alpha}_k = \frac{k\times 2\pi}{K}, \forall k \in [K]$ be a collection of points on the unit circle that is distributed uniformly. Moreover, define $\{\widehat{\theta}_k\}_{k\in[K]}$ as

$$\widehat{\theta}_k = \begin{cases} \widehat{\alpha}_{k+1} - \widehat{\alpha}_k, & \forall k \in [K-1] \\ \widehat{\alpha}_1 - \widehat{\alpha}_K + 2\pi, & k = K. \end{cases} \tag{C.40}$$

Since $\boldsymbol{W}^\star$ is a Softmax Code, we have

$$\rho_{\text{one-vs-rest}}(\boldsymbol{W}^\star) \geq \rho_{\text{one-vs-rest}}(\widehat{\boldsymbol{W}}), \tag{C.41}$$

which implies, using the definition of $\rho_{\text{one-vs-rest}}()$,

$$\min_{k\in[K]} \text{dist}(\boldsymbol{w}_k^\star, \{\boldsymbol{w}_j^\star\}_{j\in[K]\setminus k}) \geq \min_{k\in[K]} \text{dist}(\widehat{\boldsymbol{w}}_k, \{\widehat{\boldsymbol{w}}_j\}_{j\in[K]\setminus k}). \tag{C.42}$$

If there exists a $\bar{k} \in [K]$ such that $\theta_{\bar{k}}^\star + \theta_{\bar{k}+1}^\star < 2 \times \frac{2\pi}{K}$, by noting that $\widehat{\theta}_{\bar{k}} + \widehat{\theta}_{\bar{k}+1} = 2 \times \frac{2\pi}{K}$, it is easy to see geometrically that

$$\text{dist}(\boldsymbol{w}_{\bar{k}}^\star, \{\boldsymbol{w}_j^\star\}_{j\in[K]\setminus \bar{k}}) < \text{dist}(\widehat{\boldsymbol{w}}_{\bar{k}}, \{\widehat{\boldsymbol{w}}_j\}_{j\in[K]\setminus \bar{k}}) = \min_{k\in[K]} \text{dist}(\widehat{\boldsymbol{w}}_k, \{\widehat{\boldsymbol{w}}_j\}_{j\in[K]\setminus k}),$$

where the last equality follows from the fact that $\{\widehat{\boldsymbol{w}}_k\}_{k\in[K]}$ are uniformly distributed. This result contradicts (C.42), which implies that (C.39) holds.

Taking the summation on both sizes of (C.39) over all $k \in [K]$ and divide both sides by 2, we obtain

$$\sum_{k\in[K]} \theta_k^* \geq 2\pi. \tag{C.43}$$

Comparing this with (C.38), we obtain that the inequality in (C.39) holds with equality for all $k \in [K]$, that is,

$$\theta_k^\star + \theta_{k+1}^\star = 2 \times \frac{2\pi}{K}, \quad \forall k \in [K]. \tag{C.44}$$

If there exists a $\bar{k} \in [K]$ such that $\theta_{\bar{k}}^\star \neq \theta_{\bar{k}+1}^\star$, then it is easy to see geometrically that

$$\text{dist}(\boldsymbol{w}_{\bar{k}}^\star, \{\boldsymbol{w}_j^\star\}_{j\in[K]\setminus \bar{k}}) < \text{dist}(\widehat{\boldsymbol{w}}_{\bar{k}}, \{\widehat{\boldsymbol{w}}_j\}_{j\in[K]\setminus \bar{k}}) = \min_{k\in[K]} \text{dist}(\widehat{\boldsymbol{w}}_k, \{\widehat{\boldsymbol{w}}_j\}_{j\in[K]\setminus k}),$$

which contradicts (C.42). Hence, it follows that $\theta_k^\star = \frac{2\pi}{K}$ for all $k \in [K]$.

- **$K \leq d+1$: $\{\boldsymbol{w}_k^\star\}$ forms a simplex ETF.**

  We first consider optimal configuration of $K$ unit-length vectors $\boldsymbol{u}_1, \ldots, \boldsymbol{u}_K$. Note that

$$0 \leq \left\| \sum_{k=1}^K \boldsymbol{u}_k \right\|_2^2 = \sum_k \sum_{k'} \langle \boldsymbol{u}_k, \boldsymbol{u}_{k'} \rangle \leq K + K(K-1) \max_{k\neq k'} \langle \boldsymbol{u}_k, \boldsymbol{u}_{k'} \rangle,$$

  where the first inequality achieves equality only when $\sum_{k=1}^K \boldsymbol{u}_k = 0$ and the second inequality becomes equality only when $\langle \boldsymbol{u}_k, \boldsymbol{u}_{k'} \rangle = -\frac{1}{K-1}$ for any $k \neq k'$. These two conditions mean that $\boldsymbol{u}_1, \ldots, \boldsymbol{u}_K$ form a simplex ETF. The above equation further impleis that

$$\max_{k\neq k'} \langle \boldsymbol{u}_k, \boldsymbol{u}_{k'} \rangle \geq -\frac{1}{K-1}, \quad \forall \boldsymbol{u}_1, \ldots, \boldsymbol{u}_K \in \mathbb{S}^{d-1}, \tag{C.45}$$

  and the equality holds only when $\boldsymbol{u}_1, \ldots, \boldsymbol{u}_K$ form a simplex ETF.

  We will also need the following result:

$$\frac{1}{2} \sum_{k\neq k'} \|\boldsymbol{w}_k - \boldsymbol{w}_{k'}\|^2 = K^2 - \left\| \sum_k \boldsymbol{u}_k \right\|^2 \leq K^2, \tag{C.46}$$

where the last inequality becomes equality when $\sum_k \boldsymbol{u}_k = 0$.

We now prove the form of the optimal Softmax Code for $K \leq d + 1$. Noting the equivalence between Softmax Code and the HardMax problem as proved in Theorem 3.2, we will analyze the HardMax problem for this case. Specifically, note that

$$K(K-1)\max_{k \neq k'}\langle \boldsymbol{w}_{k'} - \boldsymbol{w}_k, \boldsymbol{h}_k \rangle \geq \sum_{k \neq k'}\langle \boldsymbol{w}_{k'} - \boldsymbol{w}_k, \boldsymbol{h}_k \rangle = \frac{1}{2}\sum_{k \neq k'}\langle \boldsymbol{w}_{k'} - \boldsymbol{w}_k, \boldsymbol{h}_k - \boldsymbol{h}_{k'} \rangle$$

$$\geq -\frac{1}{4}\sum_{k \neq k'}\|\boldsymbol{w}_k - \boldsymbol{w}_{k'}\|^2 - \frac{1}{4}\sum_{k \neq k'}\|\boldsymbol{h}_k - \boldsymbol{h}_{k'}\|^2 \geq -K^2,$$

where the first inequality achieves equality only when $\langle \boldsymbol{w}_{k'} - \boldsymbol{w}_k, \boldsymbol{h}_k \rangle = \langle \boldsymbol{w}_{j'} - \boldsymbol{w}_j, \boldsymbol{h}_j \rangle$ for any $k' \neq k, j' \neq j$, the second inequality follows from the Cauchy–Schwarz inequality and acheives inequality only when $\boldsymbol{w}_{k'} - \boldsymbol{w}_k = \boldsymbol{h}_{k'} - \boldsymbol{h}_k$ for any $k' \neq k$, and the third inequality follows from (C.46) and achieves equality only when $\sum_k \boldsymbol{w}_k = \sum_k \boldsymbol{h}_k = 0$. Assuming all these conditions hold, then $\langle \boldsymbol{w}_{k'} - \boldsymbol{w}_k, \boldsymbol{h}_k \rangle = -\frac{K}{K-1}$, which together with the requirement $\boldsymbol{w}_{k'} - \boldsymbol{w}_k = \boldsymbol{h}_{k'} - \boldsymbol{h}_k$ implies that

$$\langle \boldsymbol{h}_{k'} - \boldsymbol{h}_k, \boldsymbol{h}_k \rangle = -\frac{K}{K-1}, \quad \Rightarrow \quad \langle \boldsymbol{h}_{k'}, \boldsymbol{h}_k \rangle = -\frac{1}{K-1}, \forall k \neq k',$$

which holds only when $\boldsymbol{H}$ forms a simplex ETF according to the derivation for (C.45). Using the condition $\boldsymbol{w}_{k'} - \boldsymbol{w}_k = \boldsymbol{h}_{k'} - \boldsymbol{h}_k$ which indicates $\langle \boldsymbol{w}_{k'}, \boldsymbol{w}_k \rangle = \langle \boldsymbol{h}_{k'}, \boldsymbol{h}_k \rangle$, we can obtain that $\boldsymbol{W}$ is also a simplex ETF, which completes the proof.

- **$d+1 < K \leq 2d$: $\rho_{\text{one-vs-rest}}(\boldsymbol{W}^\star) = 1$ which can be achieved when $\{\boldsymbol{w}_k^\star\}$ are some vertices of a cross-polytope.**

  We first present and prove the following result that establishes an upper bound for $\rho_{\text{one-vs-rest}}(\boldsymbol{W})$ when $K \geq d + 2$.

  **Lemma C.9.** *Suppose $K \geq d+2$, then for any $\boldsymbol{W} \in \mathcal{O}\mathrm{B}(d, K)$, it holds that $\rho_{\text{one-vs-rest}}(\boldsymbol{W}) \leq 1$, with equality only if $\boldsymbol{0} \in \mathrm{conv}(\boldsymbol{W})$, where $\mathrm{conv}(\boldsymbol{W})$ is the convex hull of $\{\boldsymbol{w}_j\}_{j \in [K]}$.*

  *Proof of Lemma C.9.* Let $\boldsymbol{v}$ denote the (unique) point in $\mathrm{conv}(\boldsymbol{W})$ of minimum $l_2$ norm. By Carathéodory's theorem, $\boldsymbol{v}$ resides in the convex hull of $d + 1$ of the points in $\boldsymbol{W}$. Since $K \geq d + 2$ by assumption, there exists $k \in [K]$ such that $\boldsymbol{v} \in \mathrm{conv}(\{\boldsymbol{w}_j\}_{j \in [K] \setminus k})$ where $\boldsymbol{v}$ is the projection of $\boldsymbol{w}_k$ to $\mathrm{conv}(\{\boldsymbol{w}_j\}_{j \in [K] \setminus k})$. The result follows by noting the following two cases.

  - **Case I: $\boldsymbol{v} = \boldsymbol{0}$.** Then $\rho(\boldsymbol{W}) \leq \mathrm{dist}(\boldsymbol{w}_k, \mathrm{conv}(\{\boldsymbol{w}_j\}_{j \in [K] \setminus k})) \leq \|\boldsymbol{w}_k - \boldsymbol{v}\|_2 = \|\boldsymbol{w}_k\| = 1$.
  - **Case II: $\boldsymbol{v} \neq \boldsymbol{0}$.** Since $\boldsymbol{v}$ is the projection of $\boldsymbol{0}$ onto $\mathrm{conv}(\boldsymbol{W})$, the supporting hyperplane at $\boldsymbol{v}$ gives $\langle \boldsymbol{x}, \boldsymbol{v} \rangle \geq \|\boldsymbol{v}\|_2^2$ for every $\boldsymbol{x} \in \mathrm{conv}(\boldsymbol{W})$. In particular, taking $\boldsymbol{x} = \boldsymbol{w}_k$ implies

    $$\|\boldsymbol{w}_k - \boldsymbol{v}\|^2 = \|\boldsymbol{w}_k\|^2 - 2\langle \boldsymbol{w}_k, \boldsymbol{v} \rangle + \|\boldsymbol{v}\|^2 \leq 1 - \|\boldsymbol{v}\|^2 < 1,$$

    and so

    $$\rho_{\text{one-vs-rest}}(\boldsymbol{W}) \leq \mathrm{dist}(\boldsymbol{w}_k, \mathrm{conv}(\{\boldsymbol{w}_j\}_{j \in [K] \setminus k})) \leq \|\boldsymbol{w}_k - \boldsymbol{v}\| < 1.$$

    $\square$

According to Lemma C.9, when $K \geq d + 2$, for any $\boldsymbol{W} \in \mathcal{O}\mathrm{B}(d, K)$, it holds that $\rho_{\text{one-vs-rest}}(\boldsymbol{W}) \leq 1$. Moreover, when $d + 2 \leq K \leq 2d$, we can verify that any sphere code $\boldsymbol{W}$ that achieves equality in Rankin's orthoplex bound (Fickus et al., 2017) $\max_{k \neq j}\langle \boldsymbol{w}_k, \boldsymbol{w}_j \rangle \geq 0$ is a softmax code. In particular, for each $k$, the point $\{\boldsymbol{w}_j\}_{j \in [K] \setminus k}$ necessarily reside in the half space $H_k = \{\boldsymbol{w} : \langle \boldsymbol{w}, \boldsymbol{w}_k \rangle \leq 0\}$, and so

$$\mathrm{dist}(\boldsymbol{w}_k, \mathrm{conv}(\{\boldsymbol{w}_j\}_{j \in [K] \setminus k})) \geq \mathrm{dist}(\boldsymbol{w}_k, H_k) = 1.$$

By minimizing over $k \in [K]$, it follows that $\rho_{\text{one-vs-rest}}(\boldsymbol{W}) \geq 1$. The previous conclusion ($\rho_{\text{one-vs-rest}}(\boldsymbol{W}) \leq 1$ always hols when $K \geq d + 2$) implies that $\boldsymbol{W}$ is a softmax code.

Thus, $\boldsymbol{W}^\star = \{\boldsymbol{w}_k^\star\}$ as some vertices of a cross-polytope is a Softmax Code. In addition, since $\boldsymbol{W}^\star = \{\boldsymbol{w}_k^\star\}$ as some vertices of a cross-polytope and $K \geq d + 2$, we have $0 \in \mathrm{conv}(\{\boldsymbol{w}_j^\star\}_{j \in [K] \boldsymbol{k}})$ for any $k$, and hence it also holds that $\mathrm{dist}(\boldsymbol{w}_k^\star, \{\boldsymbol{w}_j^\star\}_{j \in [K] \boldsymbol{k}}) = 1$. Thus, there is no rattler for this case.

□

### C.5 PROOF OF THEOREM 3.5

**Theorem C.10** ($\mathcal{GNC}1$)**.** *Let* $(\boldsymbol{W}^\star, \boldsymbol{H}^\star)$ *be an optimal solution to* (5)*. For all* $k$ *that is not a rattler of* $\boldsymbol{W}^\star$*, it holds that*

$$\overline{\boldsymbol{h}_k^\star} \doteq \boldsymbol{h}_{k,1}^\star = \cdots = \boldsymbol{h}_{k,n}^\star = \mathcal{P}_{\mathbb{S}^{d-1}} \left( \boldsymbol{w}_k^\star - \mathcal{P}_{\{\boldsymbol{w}_j^\star\}_{j \in [K] \setminus k}}(\boldsymbol{w}_k^\star) \right). \tag{C.47}$$

*Proof.* Let $\bar{k}$ be any non-rattler of $\boldsymbol{W}^*$. We have

$$
\begin{aligned}
-\rho_{\text{one-vs-rest}}(\boldsymbol{W}^*) &= -\min_{k \in [K]} \text{dist}(\boldsymbol{w}_k^*, \{\boldsymbol{w}_j^*\}_{j \in [K] \setminus k}) && (\text{Definition of } \rho_{\text{one-vs-rest}}()) \\
&= -\text{dist}(\boldsymbol{w}_{\bar{k}}^*, \{\boldsymbol{w}_j^*\}_{j \in [K] \setminus \bar{k}}) && (\text{Definition of rattler}) \\
&= \min_{\boldsymbol{h} \in \mathbb{S}^{d-1}} \max_{k' \neq \bar{k}} \langle \boldsymbol{w}_{k'}^* - \boldsymbol{w}_{\bar{k}}^*, \boldsymbol{h} \rangle && (\text{Lemma C.4}) \\
&\leq \max_{i \in [n]} \max_{k' \neq \bar{k}} \langle \boldsymbol{w}_{k'}^* - \boldsymbol{w}_{\bar{k}}^*, \boldsymbol{h}_{\bar{k},i}^* \rangle && (\text{Property of max}) \\
&\leq \max_{i \in [n]} \max_{k \in [K]} \max_{k' \neq k} \langle \boldsymbol{w}_{k'}^* - \boldsymbol{w}_k^*, \boldsymbol{h}_{k,i}^* \rangle && (\text{Property of max}) \\
&= \mathcal{L}_{\text{HardMax}}(\boldsymbol{W}^*, \boldsymbol{H}^*) && (\text{Definition of } \mathcal{L}_{\text{HardMax}}) \\
&= -\rho_{\text{one-vs-rest}}(\boldsymbol{W}^{\text{SC}}) && (\text{Eq. (C.32)}) \\
&= -\rho_{\text{one-vs-rest}}(\boldsymbol{W}^*) && (\text{Theorem 3.2})
\end{aligned}
\tag{C.48}
$$

Since the first and last expressions are identical, all inequalities holds with equality. Hence

$$\min_{\boldsymbol{h} \in \mathbb{S}^{d-1}} \max_{k' \neq \bar{k}} \langle \boldsymbol{w}_{k'}^* - \boldsymbol{w}_{\bar{k}}^*, \boldsymbol{h} \rangle = \max_{i \in [n]} \max_{k' \neq \bar{k}} \langle \boldsymbol{w}_{k'}^* - \boldsymbol{w}_{\bar{k}}^*, \boldsymbol{h}_{\bar{k},i}^* \rangle. \tag{C.49}$$

By Lemma C.4, the optimal $\boldsymbol{h}$ to the optimization problem on the left is unique. Hence, all $\boldsymbol{h}_{\bar{k},i}^*, i \in [n]$ must be equal. This concludes the proof.

□

### C.6 PROOF OF THEOREM 3.7

*Proof.* ( $\implies$ ) Assume that any $(\boldsymbol{W}^\star, \boldsymbol{H}^\star) \in \arg\min_{\boldsymbol{W} \in \mathcal{OB}(d,K), \boldsymbol{H} \in \mathcal{OB}(d,nK)} \mathcal{L}_{\text{HardMax}}(\boldsymbol{W}, \boldsymbol{H})$ satisfies $\boldsymbol{h}_{k,i}^\star = \boldsymbol{w}_k^\star, \forall i \in [n], \forall k \in [K]$. We show that the Tammes problem and Softmax code are equivalent. This can be established trivially from the following two claims, namely,

$$\underset{\boldsymbol{W} \in \mathcal{OB}(d,K)}{\arg\min} \underset{\boldsymbol{H} \in \mathcal{OB}(d,nK)}{\min} \mathcal{L}_{\text{HardMax}}(\boldsymbol{W}, \boldsymbol{H}) = \underset{\boldsymbol{W} \in \mathcal{OB}(d,K)}{\arg\max} \rho_{\text{one-vs-rest}}(\boldsymbol{W}), \tag{C.50}$$

and

$$\underset{\boldsymbol{W} \in \mathcal{OB}(d,K)}{\arg\min} \underset{\boldsymbol{H} \in \mathcal{OB}(d,nK)}{\min} \mathcal{L}_{\text{HardMax}}(\boldsymbol{W}, \boldsymbol{H}) = \underset{\boldsymbol{W} \in \mathcal{OB}(d,K)}{\arg\max} \rho_{\text{one-vs-one}}(\boldsymbol{W}). \tag{C.51}$$

In the rest of the proof we show that the claims (C.50) and (C.51) hold true.

We first establish (C.50). From Theorem 3.2, any solutions to the HardMax problem is a Softmax Code, i.e.,

$$\underset{\boldsymbol{W} \in \mathcal{OB}(d,K)}{\arg\min} \underset{\boldsymbol{H} \in \mathcal{OB}(d,nK)}{\min} \mathcal{L}_{\text{HardMax}}(\boldsymbol{W}, \boldsymbol{H}) \subseteq \underset{\boldsymbol{W} \in \mathcal{OB}(d,K)}{\arg\max} \rho_{\text{one-vs-rest}}(\boldsymbol{W}). \tag{C.52}$$

Conversely, from Theorem C.7, any Softmax Code must also be a solution to the HardMax problem, i.e.,

$$\underset{\boldsymbol{W} \in \mathcal{OB}(d,K)}{\arg\max} \rho_{\text{one-vs-rest}}(\boldsymbol{W}) \subseteq \underset{\boldsymbol{W} \in \mathcal{OB}(d,K)}{\arg\min} \underset{\boldsymbol{H} \in \mathcal{OB}(d,nK)}{\min} \mathcal{L}_{\text{HardMax}}(\boldsymbol{W}, \boldsymbol{H}) \tag{C.53}$$

Combining the above two relations we readily obtain (C.50).

We now establish (C.51). Let $(\boldsymbol{W}^\star, \boldsymbol{H}^\star) \in \arg\min_{\boldsymbol{W} \in \mathcal{O}\mathrm{B}(d,K), \boldsymbol{H} \in \mathcal{O}\mathrm{B}(d,nK)} \mathcal{L}_{\mathrm{HardMax}}(\boldsymbol{W}, \boldsymbol{H})$ be any solution to the HardMax problem, which by our assumption satisfies $\boldsymbol{h}_{k,i}^\star = \boldsymbol{w}_k^*, \forall i \in [n], \forall k \in [K]$. Using this condition and plugging in the definition of the HardMax function we obtain

$$\boldsymbol{W}^* \in \underset{\boldsymbol{W} \in \mathcal{O}\mathrm{B}(d,K)}{\arg\min} \max_{k \in [K]} \max_{k' \neq k} \langle \boldsymbol{w}_{k'} - \boldsymbol{w}_k, \boldsymbol{w}_k \rangle = \underset{\boldsymbol{W} \in \mathcal{O}\mathrm{B}(d,K)}{\arg\min} \max_{k \in [K]} \max_{k' \neq k} \langle \boldsymbol{w}_{k'}, \boldsymbol{w}_k \rangle = \underset{\boldsymbol{W} \in \mathcal{O}\mathrm{B}(d,K)}{\arg\max} \rho_{\text{one-vs-one}}(\boldsymbol{W}),$$
(C.54)

where the first equality uses the fact that $\boldsymbol{w}_k$ has unit $\ell_2$ norm for all $k \in [K]$, and the second equality follows trivially from the definition of $\rho_{\text{one-vs-one}}()$. This implies

$$\underset{\boldsymbol{W} \in \mathcal{O}\mathrm{B}(d,K)}{\arg\min} \min_{\boldsymbol{H} \in \mathcal{O}\mathrm{B}(d,nK)} \mathcal{L}_{\mathrm{HardMax}}(\boldsymbol{W}, \boldsymbol{H}) \subseteq \underset{\boldsymbol{W} \in \mathcal{O}\mathrm{B}(d,K)}{\arg\max} \rho_{\text{one-vs-one}}(\boldsymbol{W}). \quad (C.55)$$

We argue that the converse of the set inclusion above also holds. To see that, let

$$\boldsymbol{W}^{\mathrm{Tammes}} \in \underset{\boldsymbol{W} \in \mathcal{O}\mathrm{B}(d,K)}{\arg\max} \rho_{\text{one-vs-one}}(\boldsymbol{W})$$

be any solution to the Tammes problem, and let $\boldsymbol{H}^{\mathrm{Tammes}} \in \mathcal{O}\mathrm{B}(d, nK)$ be such that $\boldsymbol{h}_{k,i}^{\mathrm{Tammes}} = \boldsymbol{w}_k^{\mathrm{Tammes}}, \forall i \in [n], \forall k \in [K]$. We have

$$\begin{aligned}
\mathcal{L}_{\mathrm{HardMax}}(\boldsymbol{W}^{\mathrm{Tammes}}, \boldsymbol{H}^{\mathrm{Tammes}}) &= \max_{k \in [K]} \max_{i \in [n]} \max_{k' \neq k} \langle \boldsymbol{w}_{k'}^{\mathrm{Tammes}} - \boldsymbol{w}_k^{\mathrm{Tammes}}, \boldsymbol{h}_{k,i}^{\mathrm{Tammes}} \rangle && \text{(Eq. (5))} \\
&= \max_{k \in [K]} \max_{k' \neq k} \langle \boldsymbol{w}_{k'}^{\mathrm{Tammes}} - \boldsymbol{w}_k^{\mathrm{Tammes}}, \boldsymbol{w}_k^{\mathrm{Tammes}} \rangle && \text{(Using } \boldsymbol{h}_{k,i}^{\mathrm{Tammes}} = \boldsymbol{w}_k^{\mathrm{Tammes}}) \\
&= \max_{k \in [K]} \max_{k' \neq k} \langle \boldsymbol{w}_{k'}^{\mathrm{Tammes}}, \boldsymbol{w}_k^{\mathrm{Tammes}} \rangle - 1 && \text{(Using } \|\boldsymbol{w}_k^{\mathrm{Tammes}}\|_2 = 1) \\
&= \min_{\boldsymbol{W} \in \mathcal{O}\mathrm{B}(d,K)} \max_{k \in [K]} \max_{k' \neq k} \langle \boldsymbol{w}_{k'}, \boldsymbol{w}_k \rangle - 1 && \text{(Definition of } \boldsymbol{W}^{\mathrm{Tammes}}) \\
&= \max_{k \in [K]} \max_{k' \neq k} \langle \boldsymbol{w}_{k'}^\star, \boldsymbol{w}_k^\star \rangle - 1 && \text{(Eq. (C.54))} \\
&= \max_{k \in [K]} \max_{k' \neq k} \langle \boldsymbol{w}_{k'}^\star - \boldsymbol{w}_k^\star, \boldsymbol{w}_k^\star \rangle && \text{(Using } \|\boldsymbol{w}_k^\star\|_2 = 1) \\
&= \max_{k \in [K]} \max_{i \in [n]} \max_{k' \neq k} \langle \boldsymbol{w}_{k'}^\star - \boldsymbol{w}_k^\star, \boldsymbol{h}_{k,i}^\star \rangle && \text{(Using } \boldsymbol{h}_{k,i}^\star = \boldsymbol{w}_k^\star) \\
&= \mathcal{L}_{\mathrm{HardMax}}(\boldsymbol{W}^\star, \boldsymbol{H}^\star).
\end{aligned}$$
(C.56)

This implies that $(\boldsymbol{W}^{\mathrm{Tammes}}, \boldsymbol{H}^{\mathrm{Tammes}})$ is also a solution to the HardMax problem. Hence,

$$\underset{\boldsymbol{W} \in \mathcal{O}\mathrm{B}(d,K)}{\arg\max} \rho_{\text{one-vs-one}}(\boldsymbol{W}) \subseteq \underset{\boldsymbol{W} \in \mathcal{O}\mathrm{B}(d,K)}{\arg\min} \min_{\boldsymbol{H} \in \mathcal{O}\mathrm{B}(d,nK)} \mathcal{L}_{\mathrm{HardMax}}(\boldsymbol{W}, \boldsymbol{H}). \quad (C.57)$$

Combining (C.55) and (C.57) we obtain (C.51).

($\Longleftarrow$) Assume that the Tammes problem and the Softmax Codes are equivalent. Take any optimal solution $(\boldsymbol{W}^\star, \boldsymbol{H}^\star)$ to (5), i.e.,

$$\begin{aligned}
(\boldsymbol{W}^\star, \boldsymbol{H}^\star) &= \underset{\boldsymbol{W} \in \mathcal{O}\mathrm{B}(d,K), \boldsymbol{H} \in \mathcal{O}\mathrm{B}(d,nK)}{\arg\max} \mathcal{L}_{\mathrm{HardMax}}(\boldsymbol{W}, \boldsymbol{H}) \\
&= \underset{\boldsymbol{W} \in \mathcal{O}\mathrm{B}(d,K), \boldsymbol{H} \in \mathcal{O}\mathrm{B}(d,nK)}{\arg\max} \max_{k \in [K]} \max_{i \in [n]} \max_{k' \neq k} \langle \boldsymbol{w}_{k'} - \boldsymbol{w}_k, \boldsymbol{h}_{k,i} \rangle. \quad (C.58)
\end{aligned}$$

We show that $\boldsymbol{h}_{k,i}^\star = \boldsymbol{w}_k^*, \forall i \in [n], \forall k \in [K]$.

From Theorem 3.2, $\boldsymbol{W}^\star$ is a Softmax Code, i.e.,

$$\boldsymbol{W}^* = \underset{\boldsymbol{W} \in \mathcal{O}\mathrm{B}(d,K)}{\arg\max} \min_{k \in [K]} \mathrm{dist}\left(\boldsymbol{w}_k^\star, \{\boldsymbol{w}_j^\star\}_{j \in [K] \setminus k}\right) = \underset{\boldsymbol{W} \in \mathcal{O}\mathrm{B}(d,K)}{\arg\min} \max_{k \in [K]} \min_{\boldsymbol{h}_k \in \mathbb{S}^{d-1}} \max_{k' \neq k} \langle \boldsymbol{w}_{k'} - \boldsymbol{w}_k, \boldsymbol{h}_k \rangle,$$
(C.59)

where the second equality follows from Lemma C.4. By the assumption that Softmax Code has no rattler, we know that

$$\min_{\boldsymbol{h}_k \in \mathbb{S}^{d-1}} \max_{k' \neq k} \langle \boldsymbol{w}_{k'}^\star - \boldsymbol{w}_k^\star, \boldsymbol{h}_k \rangle \quad (C.60)$$

is independent of $k \in [K]$. We now use this result to show that, for any $\bar{k} \in [K]$ and $\bar{i} \in [n]$ the following result holds:

$$h_{\bar{k},\bar{i}}^* \in \underset{h_{\bar{k}} \in \mathbb{S}^{d-1}}{\arg\min} \max_{k' \neq \bar{k}} \langle w_{k'}^\star - w_{\bar{k}}^\star, h_{\bar{k}} \rangle. \tag{C.61}$$

To prove (C.61) by constructing a contradition, we assume that (C.61) does not hold, i.e.,

$$h_{\bar{k},\bar{i}}^* \notin \underset{h_{\bar{k}} \in \mathbb{S}^{d-1}}{\arg\min} \max_{k' \neq \bar{k}} \langle w_{k'}^\star - w_{\bar{k}}^\star, h_{\bar{k}} \rangle. \tag{C.62}$$

Using (C.62) we have

$$\begin{aligned}
\mathcal{L}_{\text{HardMax}}(W^\star, H^\star) &= \max_{k \in [K]} \max_{i \in [n]} \max_{k' \neq k} \langle w_{k'}^\star - w_k^\star, h_{k,i}^\star \rangle && \text{(Definition of } \mathcal{L}_{\text{HardMax}}) \\
&\geq \max_{k' \neq \bar{k}} \langle w_{k'}^\star - w_{\bar{k}}^\star, h_{\bar{k},\bar{i}}^\star \rangle && \text{(Property of max)} \\
&> \min_{h_{\bar{k}} \in \mathbb{S}^{d-1}} \max_{k' \neq \bar{k}} \langle w_{k'}^\star - w_{\bar{k}}^\star, h_{\bar{k}} \rangle && \text{(Eq. (C.62))} \\
&= \min_{h_k \in \mathbb{S}^{d-1}} \max_{k' \neq k} \langle w_{k'}^\star - w_k^\star, h_k \rangle, \forall k \in [K] && \text{(Eq. (C.60))} \\
&= \max_{k \in [K]} \max_{i \in [n]} \min_{h_{k,i} \in \mathbb{S}^{d-1}} \max_{k' \neq k} \langle w_{k'}^\star - w_k^\star, h_{k,i} \rangle \\
&= \min_{H \in \mathbb{S}^{d-1}} \max_{k \in [K]} \max_{i \in [n]} \max_{k' \neq k} \langle w_{k'}^\star - w_k^\star, h_{k,i} \rangle \\
&\geq \min_{W \in \mathcal{OB}(d,K)} \min_{H \in \mathbb{S}^{d-1}} \max_{k \in [K]} \max_{i \in [n]} \max_{k' \neq k} \langle w_{k'} - w_k, h_{k,i} \rangle && \text{(Property of max)} \\
&= \min_{W \in \mathcal{OB}(d,K)} \min_{H \in \mathbb{S}^{d-1}} \mathcal{L}_{\text{HardMax}}(W, H) && \text{(Definition of } \mathcal{L}_{\text{HardMax}}) \\
&= \mathcal{L}_{\text{HardMax}}(W^\star, H^\star) && \text{(Eq. (C.58))}
\end{aligned}$$
$$\tag{C.63}$$

which is a contradiction. Hence, we have proved that (C.61) holds true. Now, using (C.61) we have that for any $\bar{k} \in [K]$ and $\bar{i} \in [n]$, it holds that

$$\max_{k' \neq \bar{k}} \langle w_{k'}^\star - w_{\bar{k}}^\star, h_{\bar{k},\bar{i}}^\star \rangle \leq \max_{k' \neq \bar{k}} \langle w_{k'}^\star - w_{\bar{k}}^\star, w_{\bar{k}}^\star \rangle$$

$$\implies \left( \max_{k' \neq \bar{k}} \langle w_{k'}^\star, h_{\bar{k},\bar{i}}^\star \rangle \right) - \langle w_{\bar{k}}^\star, h_{\bar{k},\bar{i}}^\star \rangle \leq \left( \max_{k' \neq \bar{k}} \langle w_{k'}^\star, w_{\bar{k}}^\star \rangle \right) - \langle w_{\bar{k}}^\star, w_{\bar{k}}^\star \rangle \tag{C.64}$$

$$\implies \left( \max_{k' \neq \bar{k}} \langle w_{k'}^\star, h_{\bar{k},\bar{i}}^\star \rangle \right) \leq \left( \max_{k' \neq \bar{k}} \langle w_{k'}^\star, w_{\bar{k}}^\star \rangle \right) + \langle w_{\bar{k}}^\star, h_{\bar{k},\bar{i}}^\star \rangle - 1.$$

On the other hand, using the result that Tammes problem and Softmax Code are equivalent, and the fact that $W^\star$ is a Softmax Code, we know that $W^\star$ is also a solution to the Tammes problem, i.e.,

$$W^\star = \underset{W \in \mathcal{OB}(d,K)}{\arg\min} \max_{k \in [K]} \max_{k' \neq k} \langle w_{k'}, w_k \rangle. \tag{C.65}$$

Hence, it must hold that for any $\bar{k} \in [K]$ and $\bar{i} \in [n]$,

$$\max_{k' \neq \bar{k}} \langle w_{k'}^\star, w_{\bar{k}}^\star \rangle \leq \max_{k' \neq \bar{k}} \langle w_{k'}^\star, h_{\bar{k},\bar{i}}^\star \rangle. \tag{C.66}$$

To see why (C.66) holds, assume for the purpose of arriving at a contradiction that it does not hold, i.e.,

$$\max_{k' \neq \bar{k}} \langle w_{k'}^\star, w_{\bar{k}}^\star \rangle > \max_{k' \neq \bar{k}} \langle w_{k'}^\star, h_{\bar{k},\bar{i}}^\star \rangle. \tag{C.67}$$

Then, consider $W^0 \in \mathcal{OB}(d,K)$ with $w_k^0 := h_{\bar{k},\bar{i}}^\star$ for $k = \bar{k}$, and $w_k^0 := w_k^\star$ otherwise. It can be verified that

$$\begin{aligned}
\max_{k \in [K]} \max_{k' \neq k} \langle w_{k'}^0, w_k^0 \rangle &= \max \left( \max_{k' \neq \bar{k}} \langle w_{k'}^0, w_{\bar{k}}^0 \rangle, \max_{\{k,k'\} \subseteq [K] \setminus \{\bar{k}\}, k' \neq k} \langle w_{k'}^0, w_k^0 \rangle \right) \\
&= \max \left( \max_{k' \neq \bar{k}} \langle w_{k'}^\star, h_{\bar{k},\bar{i}}^\star \rangle, \max_{\{k,k'\} \subseteq [K] \setminus \{\bar{k}\}, k' \neq k} \langle w_{k'}^\star, w_k^\star \rangle \right) \\
&\leq \max \left( \max_{k' \neq \bar{k}} \langle w_{k'}^\star, h_{\bar{k},\bar{i}}^\star \rangle, \max_{\{k,k'\} \subseteq [K], k' \neq k} \langle w_{k'}^\star, w_k^\star \rangle \right) \\
&\leq \max_{\{k,k'\} \subseteq [K], k' \neq k} \langle w_{k'}^\star, w_k^\star \rangle = \max_{k \in [K]} \max_{k' \neq k} \langle w_{k'}^\star, w_k^\star \rangle,
\end{aligned}$$
$$\tag{C.68}$$

where the last inequality is obtained from using (C.67). This implies, using the definition of $\boldsymbol{W}^\star$ in (C.65), that $\boldsymbol{W}^0$ is also a solution to the Tammes problem. Moreover, because of (C.67) it can be seen that $\boldsymbol{w}_{\bar{k}}^0$ is a rattler, contradicting with the assumption that the Tammes problem has no rattlers. Hence, we have proved that (C.66) holds true. Combining (C.66) with (C.64), we have

$$\max_{k'\neq\bar{k}}\langle\boldsymbol{w}_{k'}^\star,\boldsymbol{w}_{\bar{k}}^\star\rangle \leq \max_{k'\neq\bar{k}}\langle\boldsymbol{w}_{k'}^\star,\boldsymbol{h}_{\bar{k},\bar{\imath}}^\star\rangle \leq \left(\max_{k'\neq\bar{k}}\langle\boldsymbol{w}_{k'}^\star,\boldsymbol{w}_{\bar{k}}^\star\rangle\right) + \langle\boldsymbol{w}_{\bar{k}}^\star,\boldsymbol{h}_{\bar{k},\bar{\imath}}^\star\rangle - 1, \tag{C.69}$$

hence $\langle\boldsymbol{w}_{\bar{k}}^\star,\boldsymbol{h}_{\bar{k},\bar{\imath}}^\star\rangle \geq 1$. Since $\boldsymbol{w}_{\bar{k}}^\star$ and $\boldsymbol{h}_{\bar{k},\bar{\imath}}^\star$ are of unit $\ell_2$ norm, we have $\boldsymbol{h}_{\bar{k},\bar{\imath}}^\star = \boldsymbol{w}_{\bar{k}}^\star$, which concludes the proof. $\qquad\square$

## C.7   Proof of Theorem 3.8

The equivalence can be established by noting that the optimal solutions for the Tammes problem for the case $d = 2$ (Cohn, 2022) and $K \leq d + 1$ (Fickus et al., 2017) are the same as those for the Softmax codes in Theorem 3.3.

## C.8   Proof of Theorem 3.9

We first show that one-vs-rest and one-vs-one distances are mutually bounded by each other.

**Theorem C.11.** *For any classifier weights $\boldsymbol{W}$ with normalization $\|\boldsymbol{w}_k\|_2 = 1, \forall\, k \in [K]$, the one-vs-rest distance and one-vs-one distance obey the following relation:*

$$\frac{\rho_{\text{one-vs-one}}^2(\boldsymbol{W})}{2} \leq \rho_{\text{one-vs-rest}}(\boldsymbol{W}) \leq \rho_{\text{one-vs-one}}(\boldsymbol{W}). \tag{C.70}$$

*Proof.* On one hand, according to the definitions of the two margins, it is clear that $\rho_{\text{one-vs-rest}}(\boldsymbol{W}) \leq \rho_{\text{one-vs-one}}(\boldsymbol{W})$. On the other hand, according to Lemma C.4, we have

$$\rho_{\text{one-vs-rest}}(\boldsymbol{W}) = \min_k \text{dist}(\boldsymbol{w}_k, \{\boldsymbol{w}_j\}_{j\in[K]\setminus k}) = \min_k \max_{\boldsymbol{H}\in\mathcal{OB}(d,K)} \left(-\max_{j\neq k}\langle\boldsymbol{w}_j - \boldsymbol{w}_k, \boldsymbol{h}_k\rangle\right)$$

$$\geq \min_k \left(1 - \max_{j\neq k}\langle\boldsymbol{w}_j, \boldsymbol{w}_k\rangle\right) = \min_k \min_{j\neq k} \frac{\|\boldsymbol{w}_j - \boldsymbol{w}_k\|^2}{2}$$

where the first inequality follows by setting $\boldsymbol{h}_k = \boldsymbol{w}_k$. The proof is completed by noting that $\rho_{\text{one-vs-one}}^2(\boldsymbol{W}) = \min_k \min_{j\neq k} \|\boldsymbol{w}_j - \boldsymbol{w}_k\|^2$. $\qquad\square$

Combining this theorem with the existing upper bound (see Moore (1974)) and a lower bound (see Lemma C.13) on the one-vs-one distance, we obtain the following result.

**Theorem C.12.** *Assuming $K \geq \sqrt{2\pi\sqrt{e}d}$ and letting $\Gamma(\cdot)$ denote the Gamma function, we have*

$$\frac{1}{2}\left[\frac{\sqrt{\pi}}{K}\frac{\Gamma\left(\frac{d+1}{2}\right)}{\Gamma\left(\frac{d}{2}+1\right)}\right]^{\frac{2}{d-1}} \leq \max_{\boldsymbol{W}\in\mathcal{OB}(d,K)}\rho_{\text{one-vs-rest}}(\boldsymbol{W}) \leq 2\left[\frac{2\sqrt{\pi}}{K}\frac{\Gamma\left(\frac{d+1}{2}\right)}{\Gamma\left(\frac{d}{2}\right)}\right]^{\frac{1}{d-1}}. \tag{C.71}$$

Using the property $a^{1-s} < \Gamma(a+1)/\Gamma(a+s) < (a+1)^{1-s}$ for any $a > 0, s \in (0,1)$, we can simplify the bounds in (3.9) to

$$\frac{1}{2}\left[\frac{\sqrt{\pi}}{K\sqrt{\frac{d}{2}+1}}\right]^{\frac{2}{d-1}} \leq \max_{\boldsymbol{W}\in\mathcal{OB}(d,K)}\rho_{\text{one-vs-rest}}(\boldsymbol{W}) \leq 2\left[\frac{2\sqrt{\pi(\frac{d+1}{2})}}{K}\right]^{\frac{1}{d-1}},$$

**Lemma C.13.** *For any $K$ and $d$, we have*

$$\left[\frac{\sqrt{\pi}}{K}\frac{\Gamma\left(\frac{d+1}{2}\right)}{\Gamma\left(\frac{d}{2}+1\right)}\right]^{\frac{1}{d-1}} \leq \max_{\boldsymbol{W}\in\mathcal{OB}(d,K)}\rho_{\text{one-vs-one}}(\boldsymbol{W}). \tag{C.72}$$

*Proof.* We prove the theorem by constructing a $\widehat{W} \in \mathcal{O}\mathrm{B}(d, K)$ and deriving a lower bound on $\rho_{\text{one-vs-one}}(\widehat{W})$. Then, this lower bound is a lower bound for $\max_{W \in \mathcal{O}\mathrm{B}(d,K)} \rho_{\text{one-vs-one}}(W)$ as well owning to the relation

$$\rho_{\text{one-vs-one}}(\widehat{W}) \leq \max_{W \in \mathcal{O}\mathrm{B}(d,K)} \rho_{\text{one-vs-one}}(W). \tag{C.73}$$

The tightness of this lower bound naturally depends on the choice of $\widehat{W}$, which we explain below.

Instead of constructing $\widehat{W}$ directly *for a given* $K$, we first consider the construction of a $W^0(\rho_0)$ *for any given* $\rho_0 > 0$. Later, we will specify a $\rho_0$ and construct a $\widehat{W}$ from $W^0(\rho_0)$. To simplify the notation, we write $W^0(\rho_0)$ as $W^0$.

We construct $W^0$ using the following procedure from the proof of (You, 2018, Lemma 12):

- Set $w_1^0$ to be an arbitrary vector in $\mathbb{S}^{d-1}$.

- For any $k > 1$, take $w_k^0$ to be any vector in $\mathbb{S}^{d-1}$ that satisfies $\|w_k^0 - w_{k'}^0\|_2 > \rho_0$, for all $k' < k$.

- Terminate the process when no such point exists.

It is easy to see that this procedure terminates in finite number of steps; see (You, 2018, Lemma 12) for a rigorous argument. Assume that the total number of generated vectors is $K_0$. Collect these vectors into the columns of a matrix $W^0$. By construction, $W^0$ has the following property, which will be used later:

$$\rho_{\text{one-vs-one}}(W^0) > \rho_0. \tag{C.74}$$

We now derive a lower bound for $K_0$. First, note that $W^0$ provides a $\rho_0$-*covering* of the unit sphere $\mathbb{S}^{d-1}$. That is, for any $v \in \mathbb{S}^{d-1}$, there must exist a $k \in \{1, \ldots, K_0\}$ such that $\|v - w_k^0\|_2 \leq \rho_0$. Otherwise, there would exist a $w \in \mathbb{S}^{d-1}$ that satisfies $\|w - w_k^0\|_2 > \rho_0$ for all $k \leq K_0$, contradicting the termination condition in the construction of $W^0$.

Geometrically, a $\rho_0$-covering is a set of points such that, the union of Euclidean balls of radius $\rho_0$ centered at those points cover the entire unit sphere. Leveraging this interpretation, we provide a geometric method for bounding the number of points in a $\rho_0$-covering from below. Concretely, given any $w \in \mathbb{S}^{d-1}$, we denote $\mathbb{S}_{\rho_0}^{d-1}(w) = \{v \in \mathbb{S}^{d-1}, \|v - w\|_2 \leq \rho_0\}$, which is the spherical cap centered at $w$ with radius $\rho_0$. Since $W^0$ provides a $\rho_0$-covering, we have

$$\bigcup_{k=1}^{K_0} \mathbb{S}_{\rho_0}^{d-1}(w_k^0) \subseteq \mathbb{S}^{d-1} \implies \sum_{k=1}^{K_0} \sigma_{d-1}(\mathbb{S}_{\rho_0}^{d-1}(w_k^0)) \geq \sigma_{d-1}(\mathbb{S}^{d-1}), \tag{C.75}$$

where $\sigma_{d-1}$ denotes the uniform area measure on $\mathbb{S}^{d-1}$. By noting that $\sigma_{d-1}(\mathbb{S}_{\rho_0}^{d-1}(w_k^0))$ is independent of $k$, we obtain

$$K_0 \cdot \sigma_{d-1}(\mathbb{S}_{\rho_0}^{d-1}(w)) \geq \sigma_{d-1}(\mathbb{S}^{d-1}), \tag{C.76}$$

for an arbitrary choice of $w \in \mathbb{S}^{d-1}$. In above, note that the quantity $\sigma_{d-1}(\mathbb{S}_{\rho_0}^{d-1}(w))/\sigma_{d-1}(\mathbb{S}^{d-1})$ is the proportion of the area of $\mathbb{S}^{d-1}$ that lies in the spherical cap $\mathbb{S}_{\rho_0}^{d-1}(w)$. By a geometric argument, (You, 2018, Lemma 9) provides an upper bound for it which we rewrite here:

$$\frac{\sigma_{d-1}(\mathbb{S}_{\rho_0}^{d-1}(w))}{\sigma_{d-1}(\mathbb{S}^{d-1})} \leq \frac{v_{d-1}}{v_d} \left( \rho_0 \sqrt{1 - \frac{\rho_0^2}{4}} \right)^{d-1}. \tag{C.77}$$

In above, $v_d \doteq \frac{\pi^{\frac{d}{2}}}{\Gamma(\frac{d}{2}+1)}$ is the volumn of the unit Euclidean ball in $\mathbb{R}^d$. Combining (C.77) with (C.76), we obtain a lower bound on $K_0$ as

$$K_0 \geq \frac{\sigma_{d-1}(\mathbb{S}^{d-1})}{\sigma_{d-1}(\mathbb{S}_{\rho_0}^{d-1}(w))} \geq \frac{v_d}{v_{d-1}} \frac{1}{\left( \rho_0 \sqrt{1 - \frac{\rho_0^2}{4}} \right)^{d-1}} \geq \frac{v_d}{v_{d-1}} \frac{1}{\rho_0^{d-1}}. \tag{C.78}$$

We are now ready to construct a $\widehat{W} \in \mathcal{O}\mathrm{B}(d, K)$. Take

$$\rho_0 = \left( \frac{v_d}{v_{d-1} \cdot K} \right)^{\frac{1}{d-1}}. \tag{C.79}$$

By using (C.78) we have $K_0 \geq K$. Hence, we may construct $\widehat{\boldsymbol{W}}$ as the set of any $K$ distinct columns of $\boldsymbol{W}^0$. In particular, from (C.73), (C.74) and (C.79), we obtain

$$\max_{\boldsymbol{W} \in \mathcal{OB}(d,K)} \rho_{\text{one-vs-one}}(\boldsymbol{W}) \geq \rho_{\text{one-vs-one}}(\widehat{\boldsymbol{W}}) \geq \rho_{\text{one-vs-one}}(\boldsymbol{W}^0)$$

$$> \rho_0 = \left( \frac{v_d}{v_{d-1} \cdot K} \right)^{\frac{1}{d-1}} = \left[ \frac{\sqrt{\pi}}{K} \frac{\Gamma\left(\frac{d+1}{2}\right)}{\Gamma\left(\frac{d}{2}+1\right)} \right]^{\frac{1}{d-1}}, \quad \text{(C.80)}$$

where the second inequality follows trivially from the definition of $\rho_{\text{one-vs-one}}(\cdot)$. $\qquad \square$