# OpenReview forum: "Generalized Neural Collapse for a Large Number of Classes"
_ICLR.cc/2024/Conference — Submitted to ICLR 2024_

### Official Review · Reviewer_4XF7 · 2023-10-23

**Soundness:** 3 good
**Presentation:** 3 good
**Contribution:** 3 good
**Rating:** 6
**Confidence:** 3

**Summary:**

The paper introduces the concept of "generalized neural collapse", which extends the existing understanding of neural collapse in deep classification models to cases where the number of classes is much larger than the dimension of the feature space. The study reveals that when a generalized neural collapse phenomenon occurs, one-vs-rest margins is maximized. The authors provide theoretical studies for CE loss with diminishing temperate, and under some assumptions show that the optimal solutions satisfy NC1 to NC3. The authors conduct experiments to verify the theory, and propose the method of class-mean features (CMF) classifiers to reduce training memory cost.

**Strengths:**

1. This paper provides a solid theoretical study of NC for a large number of classes. Specifically, the authors reduce the problem of minimizing CE loss to the HardMax problem, and prove that NC in this setting exhibits a nice combinatoric structure called softmax code. Overall, the theory part is well motivated and clearly presented.
2. The experiments are well aligned with the theory. The proposed CMF classifier seems effective and efficient.

**Weaknesses:**

1. I still have concerns on the justifications for using a small temperature. The authors cite two papers [1,2] to support their claim. I am not convinced this is a general practice for the majority of works in this field. Also, I cannot find the exact number of $\tau=1/30$ in [1] claimed by the authors. In [2], the chosen loss function is negative cosine similarity, and I can not see how this is connected with CE with small temperature. Furthermore, I think considering only the diminishing temperature case oversimplifies the problem, since in this case, the problem intrinsically has a max-margin structure which aligns well with the desired NC properties. However, I cannot see why this is also true for general CE loss functions.
2. For GNC3, it is required that the Tammes problem and softmax codes are equivalent. The authors show that this is true for trivial setting, and conjecture it to be true for other settings. Since their separation metrics are quite different, I am not convinced that these two configurations are equivalent. As the experiments tricks in Section 5 are grounded on GNC3, I recommend the authors to make more clarifications on this matter.
3. In the related works part, the authors claim that two of the related paper focuses on networks with weight decay, which is not required in this paper. However, this paper assumes that the weight and feature vectors have unit length, which is stronger than the assumptions of using weight decay. Therefore, I can not see why this makes a big distinction.

[1] Hao Wang, Yitong Wang, Zheng Zhou, Xing Ji, Dihong Gong, Jingchao Zhou, Zhifeng Li, and Wei Liu. Cosface: Large margin cosine loss for deep face recognition.

[2] Xinlei Chen and Kaiming He. Exploring simple siamese representation learning.

**Questions:**

See the weaknesses part.

---

> ### Author Response · Authors · 2023-11-18
> **Reply to reviewer 4XF7 (part a)**
>
> Thank you for your helpful comments and insightful questions! In the following, we will answer questions one by one.
>
> **W1**: I still have concerns on ... CE loss functions.
>
> **A1**: **Face recognition**: For $tau=1/30$, we find this value in the practical implementation https://github.com/MuggleWang/CosFace_pytorch, where the inverse temperature $s=30$ is equivalent to temperature $\tau=1/30$ in our paper. To alleviate the concern of the reviewer about the general practice, we list some additional references and show the exact position of the small temperature used in these works. For face recognition, AdditiveFace[3], ArcFace[4], and NormFace[5] use the cross entropy loss with the feature and weight normalization and small temperature parameter. AdditiveFace[3] uses the temperature $\tau=1/30$ (in the first sentence of page 3); ArcFace[4] sets $\tau=1/64$ (in the second paragraph of experiment setting in section 4.2); In the NormFace[5], the authors conduct comprehensive experiments for the effect of temperature across different datasets. For example the superior performance is chosen at temperature $\tau=1/40$ on the LFW dataset in figure 8, which is also better than CE loss without any normalization.
>
> **Contrastive learning**: We indeed refer to SimCLR [6] rather than SimSiam[2]; thank you for pointing out this typo. We have corrected this in the revision. The SimCLR [6] chooses the inverse temperature between 10 and 20 (which is the inverse of our $\tau$) in table 5 to obtain the best performance. MoCo [7] sets temperature as $\tau=0.07$ according to the “technical details” part of section 3.3. In multi-domains constrative learning, ConVIRT [8] uses normalization and temperature for the CE loss to learn contrastive representations between medical image and text (see equation 2 & 3) and chooses $\tau=0.01$ (in table 4). CLIP [9] model applies normalization to both image and text embeddings to ensure consistent scales (see figure 3), and it initializes the temperature $\tau=0.07$ (in section 2.5 of page 5).
>
> **Retrieval tasks**:  The work [10] also uses the modified cross entropy loss when training dual encoders (the last paragraph in section 3) to learn the representations of query and candidates. They observe that setting a small temperature value from $\tau = 0.05$ to $0.07$ results in the best performance in table 1. Similarly, the work [11] also uses the modified cross entropy loss when training dual encoders in Equation 1 & 2 of section 2.2. It uses a temperature value of $\tau = 0.05$ for document retrieval (see the ``set up the cache loss" in https://github.com/google-research/google-research/tree/master/negative_cache.
>
> In the revision, we have incorporated these references and discussion in the appendix A.2.
>
> These applications motivate us to study the spherical constraint formulations with a small temperature, and then conduct theoretical analysis in the asymptotic case. We note that the analysis of GNC properties even for the asymptotic CE loss is non-trivial, and it provides a profound geometric interpretation through maximizing one-vs-rest margin and insights for understanding the solution structure of CE loss with small temperature.
>
> [1] Hao Wang, Yitong Wang, Zheng Zhou, Xing Ji, Dihong Gong, Jingchao Zhou, Zhifeng Li, and Wei Liu. Cosface: Large margin cosine loss for deep face recognition.
>
> [2] Xinlei Chen and Kaiming He. Exploring simple siamese representation learning.
>
> [3] Wang, Feng, et al. "Additive margin softmax for face verification." IEEE Signal Processing Letters 25.7 (2018): 926-930.
>
> [4] Deng, Jiankang, et al. "Arcface: Additive angular margin loss for deep face recognition." Proceedings of the IEEE/CVF conference on computer vision and pattern recognition. 2019.
>
> [5] Wang, Feng, et al. "Normface: L2 hypersphere embedding for face verification." Proceedings of the 25th ACM international conference on Multimedia. 2017.
>
> [6] Chen, Ting, et al. "A simple framework for contrastive learning of visual representations." International conference on machine learning. PMLR, 2020.
>
> [7] He, Kaiming, et al. "Momentum contrast for unsupervised visual representation learning." Proceedings of the IEEE/CVF conference on computer vision and pattern recognition. 2020.
>
> [8] Zhang, Yuhao, et al. "Contrastive learning of medical visual representations from paired images and text." Machine Learning for Healthcare Conference. PMLR, 2022.
>
> [9] Radford, Alec, et al. "Learning transferable visual models from natural language supervision." International conference on machine learning. PMLR, 2021.
>
> [10] Yi, Xinyang, et al. "Sampling-bias-corrected neural modeling for large corpus item recommendations." Proceedings of the 13th ACM Conference on Recommender Systems. 2019.
>
> [11] Lindgren, E., Reddi, S., Guo, R., & Kumar, S. (2021). Efficient training of retrieval models using negative cache. Advances in Neural Information Processing Systems, 34, 4134-4146.
>
> To be continued ...

---

> ### Author Response · Authors · 2023-11-18
> **Reply to reviewer 4XF7 (part b)**
>
> **W2**: For GNC3, it is required that the Tammes problem and softmax codes are equivalent. The authors show that this is true for trivial settings, and conjecture it to be true for other settings. Since their separation metrics are quite different, I am not convinced that these two configurations are equivalent. As the experiment tricks in Section 5 are grounded on GNC3, I recommend the authors to make more clarifications on this matter.
>
>
>
>
> **A2**: We first note that in experiments, GNC3 is always observed. This is the motivation for the experiment tricks (class-mean feature classifiers) in Section 5. To show GNC3, our current proof is based on the equivalence between the Tammes problem and softmax code, which is proved for certain cases. As the reviewer pointed out, the Tammes problem and softmax code involve different separation metrics, so they may be different in some cases. However, we have numerically tested all cases with $d\le 100$ in table 1 of [A], and observed equivalence between the two problems. We have incorporated this discussion in footnote 5 in the revision.
>
>
> [A]  Cohn, Henry, and Abhinav Kumar. "Universally optimal distribution of points on spheres." Journal of the American Mathematical Society 20.1 (2007): 99-148.
>
>
> **W3**: In the related works part, the authors claim that two of the related papers focus on networks with weight decay, which is not required in this paper. However, this paper assumes that the weight and feature vectors have unit length, which is stronger than the assumptions of using weight decay. Therefore, I can not see why this makes a big distinction.
>
>
> **A3**:  As shown in Appendix B.2, we observe that the two formulations lead to different geometric properties of the learned features and classifiers; in particular,  even in the case of $d = 2$, the weight decay formulation may lead to features with non-equal length and non-equal spaced distribution, and the classifiers and features are aligned only in direction but not in length.This is not the case for the spherical constraint. Consequently, as elaborated in **Q1-A1**, the spherical constraint is widely used in practice when the number of classes is large in various contexts. If we have misunderstood the reviewer's comment (as it was mentioned that “I can not see why this makes a big distinction”), we would appreciate it if the reviewer could provide further clarification on this matter. Also, we are not sure why the spherical constraint is a stronger assumption than weight decay, and we would be happy to further discuss this point if the reviewer could elaborate on this.

---

> ### Author Response · Authors · 2023-11-22
>
> Should the reviewer have any further questions or require additional insights, we are more than willing to provide detailed clarifications

---

### Official Review · Reviewer_SY44 · 2023-10-29

**Soundness:** 3 good
**Presentation:** 3 good
**Contribution:** 3 good
**Rating:** 8
**Confidence:** 5

**Summary:**

I previously reviewed this paper in NeurIPS 2023. At that time, I thought this work was the standard for acceptance, as it received a score of 76543. However, I noted that two reviewers with lower scores did not provide an objective description of the paper's contribution.  In particular, recent studies in a similar direction tend to focus on Neural Collapse when the number of categories is smaller than the feature dimension.  The contribution of this work undoubtedly broadens the research horizon and is sufficiently significant for acceptance. In the current ICLR submission, while the core content remains largely unchanged, some questions I raised during the NeurIPS 2023 preview process have been addressed, and certain revisions have been implemented in this submission. Consequently, my review remains consistent with my previous assessment.

**Strengths:**

- This paper provides a clear view of the global optimal conditions for generalized neural collapse, especially for a large number of classes, while the existing work often considers a closed-form case (i.e., a simplex of ETF) in which the number of classes is smaller than the feature dimensionality. Therefore, the contribution of the work undoubtedly broadens the horizon of Neural Collapse.
- The paper's crucial contribution lies in its demonstration that minimizing the asymptotic CE loss is equivalent to achieving within-class variability collapse and aligning with softmax codes, providing valuable insights into the underlying behavior of training neural networks.

**Weaknesses:**

- The theoretical results primarily emphasize the asymptotic CE loss rather than the original CE loss, potentially limiting the direct applicability of the findings to real-world scenarios.
- The features and class weights are constrained on the unit sphere.
- There are some minor typos, such as "where the number of classes are ... which occur" should be corrected to "where the number of classes is ... which occurs"
- Incorrect cite command in the paragraph 'Related Work' and others

**Questions:**

Please see weakness.

---

> ### Author Response · Authors · 2023-11-18
> **Reply to reviewer SY44**
>
> Thank you for your helpful comments and insightful questions! We express our gratitude to the reviewer for recognizing our work as “clear” and “valuable insight”. In the following, we will answer questions one by one.
>
> **W1**: ”The theoretical results primarily emphasize the asymptotic CE loss rather than the original CE loss, potentially limiting the direct applicability of the findings to real-world scenarios.”
>
> **A1**: We acknowledge that our theoretical results mainly focus on the asymptotic CE loss, which is our potential limitation. However, we think that there are also some advantages brought by asymptotic analysis. Unlike $K\le d$ where CE always achieves maximal margin (form a simplex ETF), when $K>d$, the geometry of the classifier and features learned by CE loss depends on the choice of temperature. As shown in Figure 2, a relatively small temperature leads to a larger margin.  In consequence, a small temperature is often used in practice, and this is the case we are interested in. Thus, it is natural to study the asymptotic case, which also provides a profound geometric interpretation through maximizing one-vs-rest margin.
>
> **W2**: “The features and class weights are constrained on the unit sphere.”
>
> **A2**: We view our emphasis on spherical constraints not as a limitation, but as a distinctive advantage. The incorporation of spherical constraints on both features and class weights has garnered considerable recognition for its demonstrated superiority in performance.  To my knowledge, the adoption of normalization for features and class weights brings at least two notable advantages within the unit sphere framework in practical contexts:
>
> **Enhanced Generalization**: Empirical investigations by Wang and Yuile [1] have underscored the effect of normalization, revealing substantial performance enhancements in face recognition (refer to Table 1 in [1]). Correspondingly, Chen et al. [2] have showcased the efficacy of explicit spherical constraints in contrastive learning, surpassing performance achieved without normalization (see Table 5 in [2]).
>
> **Mitigated Bias**: Wang et al. [3] have conducted pivotal research that elucidates how a lack of weight and feature normalization in training objectives accentuates focus on easy samples, potentially compromising performance on hard samples (as evidenced in section 3.3 of [3]). This imbalance can exacerbate bias concerns, as observed in scenarios where certain demographic groups, such as Females, Young individuals, and Black individuals, present more intricate recognition challenges (see Table 2 in [4]). Employing weight and feature normalization serves as a potent remedy to rectify these bias-related issues [3].
>
> [1] Wang, Feng, Xiang Xiang, Jian Cheng, and Alan Loddon Yuille. "Normface: L2 hypersphere embedding for face verification." In Proceedings of the 25th ACM international conference on Multimedia, pp. 1041-1049. 2017.
>
> [2] Chen, Ting, Simon Kornblith, Mohammad Norouzi, and Geoffrey Hinton. "A simple framework for contrastive learning of visual representations." In International conference on machine learning, PMLR, 2020.
>
> [3] Wang, Hao, Yitong Wang, Zheng Zhou, Xing Ji, Dihong Gong, Jingchao Zhou, Zhifeng Li, and Wei Liu. "Cosface: Large margin cosine loss for deep face recognition." In Proceedings of the IEEE conference on computer vision and pattern recognition,. 2018.
>
> [4] Klare, Brendan F., Mark J. Burge, Joshua C. Klontz, Richard W. Vorder Bruegge, and Anil K. Jain. "Face recognition performance: Role of demographic information." IEEE Transactions on information forensics and security 7, no. 6 (2012): 1789-1801.
>
> **W3-W4**: There are some minor typos, such as "where the number of classes are ... which occur" should be corrected to "where the number of classes is ... which occurs"
>
> Incorrect cite command in the paragraph 'Related Work' and others
>
> **A3-A4**: Thanks for your careful comments and advice. We have corrected the typos in the newest version.
> In terms of the cite command, we currently follow https://github.com/ICLR/Master-Template/raw/master/iclr2024.zip, which states that “when the authors or the publication are included in the sentence, the citation should not be in parenthesis using \citet { } (as in “See Hinton et al. (2006) for more information.”). Otherwise, the citation should be in parenthesis using \citep{ } (as in “Deep learning shows promise to make progress towards AI (Bengio &LeCun, 2007).”)”
> That being said, we are not very familiar with the usage and we would appreciate it if the reviewer could help make further clarification on this matter.

---

> ### Author Response · Authors · 2023-11-22
>
> Should the reviewer have any further questions or require additional insights, we are more than willing to provide detailed clarifications

---

> > ### Comment · Reviewer_SY44 · 2023-11-23
> >
> > I appreciate the authors for providing their responses. I will maintain my original score.

---

### Official Review · Reviewer_MYoA · 2023-10-30

**Soundness:** 3 good
**Presentation:** 3 good
**Contribution:** 3 good
**Rating:** 6
**Confidence:** 4

**Summary:**

The paper delves into the mathematical characterization of learned last-layer representations and classifier weights in deep classification models, known as the "Neural Collapse" (NC) phenomenon. While existing studies on NC often focus on scenarios where the number of classes is small compared to the feature space dimension, this paper extends the understanding of NC to situations where the number of classes is much larger than the feature dimension. The authors introduce the concept of "Generalized Neural Collapse" (GNC) and demonstrate that features and classifiers display a GNC phenomenon where the minimum one-vs-rest margins are maximized. The paper provides both empirical and theoretical studies to validate the occurrence of GNC in practical deep neural networks.

**Strengths:**

+ The paper is well-organized, allowing readers to easily follow the author's ideas. The writer makes sure the content is clear and simple, so readers from different backgrounds can understand the main points.
+ The author expanded the "Neural Collapse" (NC) idea to the "General Neural Collapse" (GNC), considering cases where the number of categories is greater than the feature size.

**Weaknesses:**

+ **Too strong assumption in the theoretical analysis**: $\tau \rightarrow 0$ means the norm of the feature goes to infinity, which means that a small angle between every two classes can enjoy the CE closing to $0$. Therefore, practically, GNC can not be achieved when $\tau$ is very small.
+ **Insufficient Discussions**: The Tammes problem is the limit case of Thomson-P problem [1], authors should also consider it and provide more discussions for it, especially Thomson-1 problem.

[1] Numerical Solutions of the Thomson-P Problems

**Questions:**

None

---

> ### Author Response · Authors · 2023-11-18
> **Reply to reviewer MYoA**
>
> Thank you for your helpful comments and suggestions! We appreciate that you found our work “well-organized” and acknowledge our contributions in extending NC to the general case.
>
> **W1**:  Too strong assumption in the theoretical analysis: tau→0 means the norm of the feature goes to infinity, which means that a small angle between every two classes can enjoy the CE closing to 0. Therefore, practically, GNC can not be achieved when tau is very small.
>
> **A1**: Thanks for your question. The features and classifiers are always normalized to have unit norm. Thus, when $\tau\rightarrow 0$, a pair of classes with a small angle actually would lead to large CE, instead of closing to 0. Moreover, In the revision, we have included more references and discussion of the normalization and small temperature used by the existing widespread literature in the appendix.
>
> **W2**: Insufficient Discussions: The Tammes problem is the limit case of Thomson-P problem [1], authors should also consider it and provide more discussions for it, especially Thomson-1 problem.
>
> **A2**: Note that this paper focuses on the geometry of the CE loss, which in the limit case is equivalent to the Softmax Code that concerns one-vs-rest distance. Tammes problem is introduced as it maximizes the one-vs-one distance, which is close to the one-vs-rest distance. Moreover, when GNC3 holds, the two problems are equivalent as indicated by eq. (10) and eq. (11). Thus, the Tammes problem is not the main focus of this paper. That being said, we thank the reviewer for pointing out the general Thomson problem, which has been discussed and cited in Section 2 (Interpretation of Softmax Code) in the revision.
>
> [1] Numerical Solutions of the Thomson-P Problems

---

> ### Author Response · Authors · 2023-11-22
>
> Should the reviewer have any further questions or require additional insights, we are more than willing to provide detailed clarifications

---

### Official Review · Reviewer_7nq8 · 2023-11-01

**Soundness:** 3 good
**Presentation:** 2 fair
**Contribution:** 3 good
**Rating:** 8
**Confidence:** 3

**Summary:**

The paper investigates the neural collapse geometry of CE loss with normalized features and classifiers in the case of large number of classes K>d+1. They formalize two key properties (in addition to the vanilla NC1) describing the geometry: (i) Classifiers converge to an arrangement (which they call softmax code) of K vectors on the d-dim sphere that maximizes the minimum distance of one of them to the cvx hull of the rest (ii) Embeddings align with classifiers. Empirically, they demonstrate that property (ii) holds and property (i) holds for small enough temperature parameter on the CE loss. They also show that small temperature leads to good generalization. On theory front, the authors confirm the formalization by studying the corresponding UFM with temperature tending to zero. Finally, motivated by property (ii) they propose methods that avoid training the classifier weights showing that retain good performance while being less heavy on compute.

**Strengths:**

* important problem / open question: NC geometry for large K
* technical results appear solid although I did not have time to fully check the proofs
* result is to my knowledge novel

**Weaknesses:**

* the requirement for feature/weight normalization might seem restrictive. The authors mention that this is standard practice, but looking at the three references, only one of them is on CE loss.
* the description of geometry is elegant, but is in general not explicit. Instead given as solution to a non-convex problem. It nevertheless gives an implicit way of thinking about the classifiers geometry. For d=2 it is nice to have the closed-form, but perhaps uniformity on the sphere is not very surprising (what else could it be?)
* the characterization does not specify uniquely how the classifier weights are assigned to classes. The authors discuss this under the assignment problem in Sec 4, but there does not yet appear to be a concrete answer to that

**Questions:**

Can you please elaborate on how your findings compare to the results by Liu et al (2023) and Gao et al (2023)? From the Related work part in Sec. 1, I understand you are claiming that the geometry with spherical constraints is different compared to that with weight-decay. Is this the case? If so, I believe it would be useful/interesting to elaborate on the differences.

Also wondering if you have thoughts on this: Is the use of spherical constraints more of a mathematical convenience to arrive at a more "clean" NC geometry description OR is the claim that this is a practice actually preferred over weight decay?

Thm 3.3 last statement: Is cross-polytope the only solution?

minor: "The following result shows that the requirement in the Theorem 3.5 that k is not a rattler is satisfied
in many cases"  I would say "d=2" and "K<=d+1" is *some* cases

I recommend ploting GCN1 in log scale

Can you elaborate on the numerical optimization of Softmax Code?

Computing \rho_{one-vs-rest} looks computationally expensive for large k (solving k quadratic programs). Can you please comment on that?

In addition to GNC2 metric, is it not possible to track a metric that more accurately captures the geometry? Eg. angles between classifiers. At least for cases that you know the geometry of Ws like K<2d?

The last subfigure in Fig 2 shows that generalization performance improves with decreasing temperature. Could you please comment on whether the best value achieved is also on par with the sota value for CE training without normalization and temperature scaling (which I believe is the "standard" practice?)

PS: In general, I enjoyed reading the paper and the results. I am inclined to increase my score to an 8 after the rebuttal.

---

> ### Author Response · Authors · 2023-11-18
> **Reply to reviewer 7nq8 (part a)**
>
> Thank you for your helpful comments and insightful questions! We appreciate that you found our work “important”, “solid” and “novel”. In the following, we will answer questions one by one.
>
> **W1**: “ the requirement for feature/weight normalization might seem restrictive. The authors mention that this is standard practice, but looking at the three references, only one of them is on CE loss.”
>
> **A1**: Thank you for your question. In cases where the number of classes is larger than the feature dimensions, it is common practice to add normalization and temperature to the CE loss in face recognition, information retrieval and self-supervised contrastive learning areas. Due to limited space, we only cited three representative works. In the revision, we have included more references and discussion in the appendix A.2. We will discuss the practical benefits of normalizing features and weights, and provide relevant existing literature on the topic in the **Q5-A5**.
>
> **W2**: “the description of geometry is elegant, but is in general not explicit. Instead given as solution to a non-convex problem. It nevertheless gives an implicit way of thinking about the classifiers geometry. For d=2 it is nice to have the closed-form, but perhaps uniformity on the sphere is not very surprising (what else could it be?)”
>
> **A2**: We thank the reviewer for acknowledging the description of geometry (implicitly characterized) as elegant. We think describing the geometry for any configurations of $d$ and $K$ (except for special cases, such as $K\le d+1$) in closed form is an extremely challenging problem. The closely related Tammes problem, first proposed in the 1930s and studied for nearly a century, remains an unsolved puzzle, with confirmed closed-form solutions only for certain configurations such as the cross-polytope structure when $K=2d$ and the 600-cell structure for $d=4$ and $K=120", as shown in Table 1 of [1].
>
> [1] Cohn, Henry, and Abhinav Kumar. "Universally optimal distribution of points on spheres." Journal of the American Mathematical Society 20.1 (2007): 99-148.
>
> **W3**: “the characterization does not specify uniquely how the classifier weights are assigned to classes. The authors discuss this under the assignment problem in Sec 4, but there does not yet appear to be a concrete answer to that”
>
> **A3**: We agree with the reviewer's observation that our theoretical results only define the configuration of the classifier weight and do not provide specific assignments. Since analyzing deep networks is particularly difficult due to their nonlinearity and the interplay between different layers, we adopt the unconstrained feature model that is widely used in previous studies on neural collapse, treating intermediate features as unconstrained optimization variables. However, such simplification overlooks the semantic information of the input data, resulting in unclear assignments for the classifiers. To our knowledge, it is an open problem to study neural collapse beyond the unconstrained feature model. One direction is to incorporate semantic information from the input data, but a comprehensive investigation is beyond the scope of the current work and is deferred to future work. In the revision, we have briefly mentioned this at the end of Section 4.
>
> **Q4**: “Can you please elaborate on how your findings compare to the results by Liu et al (2023) and Gao et al (2023)? From the Related work part in Sec. 1, I understand you are claiming that the geometry with spherical constraints is different compared to that with weight-decay. Is this the case? Yes If so, I believe it would be useful/interesting to elaborate on the differences.”
>
> **A4**: The reviewer is that both Liu et al (2023) and Gao et al (2023) primarily address the weight decay formulation, whereas our work focuses on the spherical constraints, and the formulations lead to different geometry of the learned classifiers and features. Due to the limited space, we present this difference in detail in Appendix B.2, but we have highlighted this in sec.1 (related work part) in the revision; thank you for the suggestion. If needed, we are willing to provide further elaboration on the distinction between our work and these two studies.
>
> To be continued ...

---

> ### Author Response · Authors · 2023-11-18
> **Reply to reviewer 7nq8 (part b)**
>
> **Q5**: “Also wondering if you have thoughts on this: Is the use of spherical constraints more of a mathematical convenience to arrive at a more "clean" NC geometry description OR is the claim that this is a practice actually preferred over weight decay?”
>
> **A5**: We consider that spherical constraints offer advantages from both perspectives. Firstly, the original CE loss is difficult to analyze due to the lack of elegant geometric properties and closed-form solutions. However, the introduction of spherical constraints allows for asymptotic analysis of the CE loss, simplifying it to the Hardmax problem. Secondly, feature/weight normalization has gained recognition for its demonstrated superiority in practical applications. We summarize the reasons and practical details in two aspects:
>
> **Enhanced Generalization**: Empirical investigations by Wang and Yuille [2] have highlighted the impact of normalization, showing significant performance improvements in face recognition (see Table 1 in [2]). Similarly, Chen et al. [3] have demonstrated the effectiveness of explicit spherical constraints in contrastive learning, surpassing performance achieved without normalization (see Table 5 in [3]).
>
> **Mitigated Bias**: Wang et al. [4] have conducted crucial research that explains how the absence of weight and feature normalization in training objectives amplifies the focus on easy samples, potentially compromising performance on difficult samples (as evidenced in section 3.3 of [4]). This imbalance can exacerbate bias concerns, particularly in scenarios where certain demographic groups, such as females, young individuals, and black individuals, present more complex recognition challenges (see Table 2 in [5]). Incorporating weight and feature normalization serves as a powerful solution to address these bias-related issues [4].
>
> [2] Wang, Feng, Xiang Xiang, Jian Cheng, and Alan Loddon Yuille. "Normface: L2 hypersphere embedding for face verification." In Proceedings of the 25th ACM international conference on Multimedia, pp. 1041-1049. 2017.
>
> [3] Chen, Ting, Simon Kornblith, Mohammad Norouzi, and Geoffrey Hinton. "A simple framework for contrastive learning of visual representations." In International conference on machine learning, PMLR, 2020.
>
> [4] Wang, Hao, Yitong Wang, Zheng Zhou, Xing Ji, Dihong Gong, Jingchao Zhou, Zhifeng Li, and Wei Liu. "Cosface: Large margin cosine loss for deep face recognition." In Proceedings of the IEEE conference on computer vision and pattern recognition,. 2018.
>
> [5] Klare, Brendan F., Mark J. Burge, Joshua C. Klontz, Richard W. Vorder Bruegge, and Anil K. Jain. "Face recognition performance: Role of demographic information." IEEE Transactions on information forensics and security 7, no. 6 (2012): 1789-1801.
>
> **Q6**: “Thm 3.3 last statement: Is cross-polytope the only solution?”
>
> **A6**: Thanks for the question. When $d+2 \le K < 2d$, subset of vertices of a cross-polytope are not the only solutions. Indeed, any sphere code that achieves equality in Rankin's orthoplex bound. To give one example, consider the case of $K = 5$ classes distributed in $d=3$-dimensional space. The weight matrix with columns [1, 0, 0], [-1/2, $\sqrt{3}/2$, 0], [-1/2, $-\sqrt{3}/2$, 0], [0, 0, 1], [0, 0, -1] (with pairwise angles being either 90 degress or 120 degrees) is a softmax code, but this is not a subset of the cross-polytope. In the revision, we have added a footnote at Thm 3.3 and updated the proof to incorporate this discussion.
>
> **Q7-Q8**: "The following result shows... not a rattler is satisfied in many cases" I would say "d=2" and "K<=d+1" is some cases.
> I recommend ploting GCN1 in log scale”
>
> **A7-A8**：Thanks for the reviewer's helpful advice.  We have incorporated the suggestion and plotted GNC1 in Fig 2 in the log scale of the revised version.
>
> **Q9-10**: “Can you elaborate on the numerical optimization of Softmax Code?” Computing $\rho_{one-vs-rest}$ looks computationally expensive for large k (solving k quadratic programs). Can you please comment on that?
>
> **A9-10**: Thank you for the great question. According to Theorem 3.2 (GNC2), solving the Softmax Codes is equivalent to solving the "HardMax" problem in eq. (5), which is an optimization over both $W$ and $H$. Thus, we optimize eq. (5) numerically using projected gradient descent. We provide the implementation of optimization of Softmax Code in the anonymous github link(https://anonymous.4open.science/r/Neural_Collapse_For_Large_Number_Of_Classes_Optimization_Softmax_Codes-4F2D/hardmax_solution.py). Likewise, when given $W$, we find the one-vs-rest distance by optimizing the “HardMax” problem over $H$. The equivalence is proved in Lemma C. 4 (eq. (C.18)) in Appendix C. Consequently, we can use the same projected gradient descent to find the one-vs-rest distance, which is more computationally efficient than solving the quadratic programs.
>
> To be continued ...

---

> ### Author Response · Authors · 2023-11-18
> **Reply to reviewer 7nq8 (part c)**
>
> **Q11**: In addition to the GNC2 metric, is it not possible to track a metric that more accurately captures the geometry? Eg. angles between classifiers. At least for cases that you know the geometry of Ws like K<2d?
>
> **A11**:  Even for $K<2d$ (except for $K\le d+1$), as per **Q6-A6**, the solution may not be unique. Thus, we use the smallest one-vs-rest distance to capture GNC2. But we would be happy to incorporate any suggestions the reviewer may have.
>
> **Q12**: “The last subfigure in Fig 2 shows that generalization performance improves with decreasing temperature. Could you please comment on whether the best value achieved is also on par with the sota value for CE training without normalization and temperature scaling (which I believe is the "standard" practice?)”
>
> **A12**: Thanks for the question and suggestion.  We have conducted a comparison with CE loss with weight decay instead of normalization and temperature scaling. The re-plotted figure 2 shows that CE with normalization and temperature achieved better performance than the weight decay. Beyond this simple illustrative case, previous studies have shown that incorporating feature/weight normalization and temperature scaling, improves performance in practical applications such as face recognition (see Table 1 in [2]) and contrastive learning (see Table 5 in [3]).
>
> [2] Wang, Feng, Xiang Xiang, Jian Cheng, and Alan Loddon Yuille. "Normface: L2 hypersphere embedding for face verification." In Proceedings of the 25th ACM international conference on Multimedia, pp. 1041-1049. 2017.
>
> [3] Chen, Ting, Simon Kornblith, Mohammad Norouzi, and Geoffrey Hinton. "A simple framework for contrastive learning of visual representations." In International conference on machine learning, PMLR, 2020.

---

> ### Author Response · Authors · 2023-11-22
>
> Should the reviewer have any further questions or require additional insights, we are more than willing to provide detailed clarifications.

---

> > ### Comment · Reviewer_7nq8 · 2023-12-04
> > **thank you for the responses**
> >
> > I thank the authors for their detailed responses. I enjoyed reading the paper and I am happy to see it published at the conference.  While the spherical constraints and vanishing temperature analysis might impose some limitations to the generality of the results, I found the approach interesting and the arguments slick. I raise my score.

---

### Meta-Review · Area_Chair_RCB3 · 2023-12-10

**Metareview:**

The paper considers the geometry of neural collapse for the case of temperature going to 0, and number of classes being potentially bigger than the dimension of the latent variables. The authors work in the "unconstrained features" setting, wherein the optimization model being discussed has both latents and classifier weights as "free" variables to be trained.

The strengths of the paper are a fairly thorough geometric analysis in the limits considered, and clean connections to the Tammes problem. Some of these connections are direct, some are gotten by noting that the variation of the problem considered (one-vs-rest) still retains some aspects of the problem (e.g. the concept of rattlers).

The weaknesses are that the assumption of the paper are fairly strong (as some reviewers noted). The "unconstrained features" setup, even though it's appeared in prior papers, is quite a blunt approximation of what might happen in practice (the features are clearly not "free" for standard neural net training). Moreover, the temperature going to 0 and normalization constraints are also quite blunt: it's common to think of the temperature as small, and the features as approximately normalized, but setting the temperature *exactly* to 0 is never done --- so to that extent, the results of the paper are only a coarse approximation of what happens during normal neural net training. Moreover, the theory doesn't suggest anything about how the geometric insights could aid neural net training --- even heuristically.

**Justification For Why Not Higher Score:**

The paper has some interesting theoretical tools developed to understand the geometry of neural collapse in the limit of low temperature, and in the setup of "unconstrained features". The proofs are elegant and there's a nice connection to classical geometric problems (the Tammes problem), but the assumptions of the theory are rather strong and coarse approximations of real-life training. This would in general be ok if the theory suggested ways to improve neural network training --- which at present it does not.

**Justification For Why Not Lower Score:**

N/A

---

### Decision · Program_Chairs · 2024-01-16

Reject